# Chemically modified CRISPR-Cas9 enables targeting of individual G-quadruplex and i-motif structures, revealing ligand-dependent transcriptional perturbation

Sabrina Pia Nuccio[1,2], Enrico Cadoni [1,3], Roxani Nikoloudaki[1], Silvia Galli[1], An-Jie Ler[1], Claudia Sanchez-Cabanillas[1,4], Thomas E. Maher[1,2], Ella Fan[1], Dilek Guneri[5], Gem Flint[1,2], Minghui Zhu[1], Ling Sum Liu[1], Christopher R. Fullenkamp[6], Zoë Waller[5], Luca Magnani[4,7], John S. Schneekloth Jr[6] & Marco Di Antonio[1,2,8] ✉

The development of selective ligands to target DNA G-quadruplexes (G4s) and i-motifs (iMs) has revealed their relevance in transcriptional regulation. However, most of these ligands are unable to target individual G4s or iMs in the genome, limiting their scope. Herein, we describe an Approach to Target Exact Nucleic Acid alternative structures (ATENA) that relies on the chemical conjugation of established G4 and iM ligands to a catalytically inactive Cas9 protein (dCas9), enabling their individual targeting in living cells. ATENA demonstrates that the selective targeting of the G4 present in the oncogene *c-MYC* leads to the suppression of transcripts regulated exclusively by one of its promoters (P1). Conversely, targeting the *c-MYC* iMs on the opposite strand leads to the selective increase of P1-driven transcripts. ATENA reveals that G4-mediated transcriptional responses are highly ligand-specific, with different ligands eliciting markedly different effects at the same G4 site. We further demonstrate that the basal expression levels of the gene targeted can be used to predict the transcriptional impact associated with G4-stabilization. Our study provides a platform for investigating G4- and iM-biology with high precision, unveiling the therapeutic relevance of individual DNA structures with selectivity.

G-quadruplex (G4) structures can promptly form within single-stranded DNA sequences rich in guanines by generating stacks of G-quartets held together by Hoogsteen hydrogen bonding that are further stabilized by coordination of potassium ions (Fig. 1a)[1]. The ability of G-rich sequences to form G4s under physiological conditions has been known for decades[2,3]. Yet, the biological relevance of these structures has been heavily disputed until recently. Over the past decade, the development of orthogonal approaches to detect and map

[1]Imperial College London, Department of Chemistry, Molecular Sciences Research Hub, 82 Wood Lane, London, UK. [2]Imperial College London, Institute of Chemical Biology, Molecular Sciences Research Hub, 82 Wood Lane, London, UK. [3]Department of Organic and Macromolecular Chemistry - Organic and Biomimetic Chemistry Research Group (OBCR), Faculty of Sciences, Campus Sterre, Krijgslaan 281, Building S4, Gent, Belgium. [4]The Breast Cancer Now Toby Robins Research Centre, The Institute of Cancer Research, 123 Old Brompton Road, London, UK. [5]UCL School of Pharmacy, 29-39 Brunswick Square, London, UK. [6]Chemical Biology Laboratory, National Cancer Institute, Frederick, MD, USA. [7]Department of Oncology and Haemato-Oncology, Università degli Studi di Milano, Milan, Italy. [8]The Francis Crick Institute, 1 Midland Road, London, UK. ✉e-mail: m.di-antonio@imperial.ac.uk

G4s has provided robust evidence to support their formation in living cells. These include immuno fluorescence[4], live-cell imaging[5,6], and genome-wide mapping strategies[7–10].

While G4 formation in cells has been validated, its active contribution in regulating biological processes is yet to be fully demonstrated. Given the high enrichment of G4s at gene promoters as measured experimentally in vitro and in cells[9,11], it has long been speculated that these structures may play an essential role in regulating gene expression[12]. Indeed, many G4-selective ligands have been developed to date, revealing that targeting G4s within gene promoters is generally associated with transcriptional suppression[13]. Since key oncogenes such as *c-MYC, KRAS,* and *BCL-2* bear a G4 motif in their promoter, applying G4 ligands for cancer intervention has been investigated[14]. However, the human genome displays more than

**Fig. 1 | Chemical labeling of dCas9-Halo for selective G4-targeting. a** Schematic representation of a G-Tetrad via Hoogsteen base pairing with central cation (K$^+$) (modified from PDB file:6W9P) (**left**) and G-tetrad stacking to form G4 structure (modified from PDB file:6W9P using Protein Imager[96]) (**right**). **b** Schematic overview of ATENA: dCas9-Halo fusion protein functionalized with chloroalkane-modified G4 ligands (created with BioRender, https://BioRender.com/hgoel9a) enables single G4-targeting through sgRNA guidance. **c** Chemical structure of chloroalkane-modified PyPDS, where n indicates the different PEG linker lengths (Cl-PyPDS$_n$). **d** Chemical structure of chloroalkane-modified PhenDC3 (Cl-PhenDC3$_n$), where n indicates the different PEG linker lengths (Cl-PhenDC3$_n$). **e** Illustration (created with BioRender, https://BioRender.com/vmzonqy) of the competition assay workflow to test the binding ability of Cl-PyPDS$_n$ probes to dCas9-Halo purified recombinant protein. **f** SDS-PAGE gel of the Cl-PyPDS$_n$ competition assay shows each sample's fluorescent level acquired in the TAMRA channel (542 nm) and the corresponding protein level (Coomassie staining): ($n = 2$). **g** Schematic representation (created with BioRender, https://BioRender.com/hirbnfn) of the dually labeled FRET oligos containing c-KIT2-G4-forming sequence bound by dCas9-Halo labeled with Cl-PyPDS$_n$ probes to study G4 stabilization with respect to the PEG linker length and sgRNA positioning. **h** ΔFRET efficiency of the decorated dCas9-PDS (with Cl-PyPDS$_2$) complex targeting c-KIT2-G4. The values indicated were extrapolated from the band intensity measured in the Cy3 and Cy5 channels (Typhoon FLA 9500). The signals in both channels were normalized for the background and the sgRNA NTC control. These normalized fluorescence values were then used to calculate the ΔFRET efficiency for each sgRNA: FRET-Efficiency ($E$)+ligand $_{sgRNAx}$ − FRET-Efficiency ($E$)-ligand $_{sgRNAx}$ ($n = 2$). Data presented are the mean of $n$ = number of independent experiments. Statistical significance was calculated using a two-tailed $t$ test in GraphPad Prism; $p$-value: ns > 0.05, *≤0.05, **≤0.01, ***≤0.001, ****≤0.0001. Source data are provided as a Source Data file.

700,000 experimentally detected G4 structures in vitro[11,15] and -10,000 G4s detected in chromatin using genomics strategies[7–10,16]—making single-G4 targeting by small molecules challenging. Indeed, most ligands recognize G4s by establishing π-π end-stacking with G-tetrads, a promiscuous interaction that hampers these molecules from displaying significant inter-G4 selectivity[17]. For example, evidence supports that the transcriptional suppression of c-MYC observed upon treatment with certain G4 ligands is an indirect consequence of global G4 stabilization rather than a specific response regulated exclusively by the G4 in the c-MYC promoter[18].

Additionally, there are experimental discrepancies between the potential endogenous biological function of G4s and what is observed when ligands bind these structures. A series of independent genomic studies consistently indicated that G4 formation is associated with active gene expression[12,19], contrasting the transcriptional repression mostly observed upon G4-ligand treatment. Similarly, recent studies have leveraged gene-editing techniques to demonstrate that the selective deletion of the G-rich sequence in the c-MYC promoter responsible for the G4 structure formation on this promoter (MYC-G4) is associated with a loss of transcriptional activity[20], further pointing to an active role of G4s in stimulating transcription. To justify the discrepancies between endogenous G4 function and the responses observed with G4-targeting small-molecules, it has been postulated that ligand binding can prevent key transcriptional factors from recognizing G4s, leading to transcriptional repression[21]. However, it remains challenging to discern whether transcriptional suppression at specific genes is caused by protein displacement at a given G4 site or is a response triggered by global G4 stabilization, as reported for c-MYC[18]. It is also often supposed that structurally different G4 ligands should elicit similar responses when targeting the same G4s, with the assumption that G4-binding proteins respond identically to a ligand-bound G4 irrespective of the ligand used. Overall, this highlights the urgent need for tools that can provide inter-G4 selectivity to widely used G4 ligands to fully underpin G4 biology and harness the therapeutic potential of these DNA secondary structures.

Alongside G4s in genomic contexts, there are complementary regions that are rich in cytosine. C-rich sequences are capable of forming i-motifs (iMs). Like G4s, these are also four-stranded structures but are stabilized by hemi-protonated, intercalated cytosine-cytosine base pairs[22,23]. The requirement for iMs to be hemi-protonated, necessitating slightly acidic conditions for formation, combined with the significant application of these structures in pH-responsive nanotechnologies[24,25], initially led to iMs not being considered biologically relevant. However, there is now substantial evidence supporting that iMs can form under physiological pH and are stabilized under conditions compatible with cellular context[26–30].

More recently, the discovery of an iM-specific antibody (iMab) has enabled iMs to be visualized in the nuclei of human cells[31]. iMab has since been used to indicate the presence of iMs in cells, including mapping of iM structures throughout the human genome[32] and others,

providing evidence that iMs and G4s are interdependent in cells[33]. Although there has been some indication that iMab may have issues with specificity[34], this antibody can be used to identify the presence of both inter and intramolecular structures[35]. Additionally, iMs have been linked to telomere maintenance and transcriptional regulation[32], with numerous examples of native proteins that are able to interact with iM structures[36–38]. In-cell NMR has also provided alternative non-antibody-based evidence of their existence in cells[30,39]. Although several small-molecule ligands have been shown to bind the i-motif-forming sequences in c-MYC[31], the exact biological responses elicited by iM ligands still remain elusive, owing to their lack of inter-iMs selectivity that confounds phenotypes arising from global versus site-specific iM targeting.

It is evident that the absence of inter-G4 or inter-iMs selectivity displayed by most of the available ligands hampers their use as therapeutic agents and their application as reliable tools to investigate the biology of these DNA structures. Hence, the development of ligands displaying preferential binding toward a specific G4 or iM structure (i.e., c-MYC) has represented a longstanding quest in this area of research. To this end, Schneekloth and co-workers recently designed a small-molecule ligand (DC-34) that displays preferential binding to the G4 in the c-MYC promoter[40]. Treatment of multiple myeloma cell lines with DC-34 revealed suppression of c-MYC transcription with minimal perturbation of other G4-bearing oncogenes, such as KRAS and BCL-2[40]. Similarly, our group and others have demonstrated that short peptides and oligonucleotides can also exhibit binding selectivity towards specific G4s[41–44]. Nevertheless, these remain isolated examples that require ligand optimization to achieve binding selectivity against any given DNA structure of interest and are not suitable for high-throughput screening of G4s or iMs function at scale. Furthermore, the highly diverse chemical and structural nature of ligands targeting G4s and iMs might represent an additional confounding factor. For example, different G4 ligands might elicit different biological responses when binding to the same G4s, leading to experimental observations that are dependent on the type of ligand used, rather than reflecting the endogenous biological function of the targeted G4.

To overcome these limitations, we have developed ATENA (Approach to Target Exact Nucleic Acid alternative structures), a CRISPR-Cas9-based platform that enables selective localization of different G4 or iM ligands in the proximity of a given DNA structure of interest (Fig. 1b). To achieve this, we have exploited HaloTag technologies to selectively install modified G4 or iM binding ligands onto Cas9 protein in living cells[45]. More specifically, we have designed a small library of established G4-ligands (pyridostatin (PDS)[46] and PhenDC3[47]) functionalized with chloroalkane side chains that ensure incorporation into a nuclease-inactive Cas9 protein fused to a HaloTag (dCas9-Halo)[48]. Similarly, we decorated one of the iM-selective peptides (Pep-RVS)[49], recently developed by the Waller group, with a chloroalkane side chain to explore selective iM targeting with ATENA. The synthesized analogs of PDS, PhenDC3, and Pep-RVS contain

different polyethylene glycol (PEG)-linkers ($n = 0, 2, 4$, Fig. 1 c, d), with the PEG chain serving to tether the DNA-binding scaffold to the HaloTag-binding chloroalkane. By fine-tuning the PEG linker length, we optimized the spacing between the DNA-binding moiety and the dCas9-Halo fusion to attain ideal G4 engagement with ATENA, using a dedicated FRET-based assay. We leveraged this knowledge to deploy ATENA in cells, achieving selective targeting of either the G4 or the iM present in the promoter of the proto-oncogene *c-MYC*. ATENA-mediated G4-targeting resulted in the reduction of *c-MYC* transcripts generated exclusively by one of the four promoters regulating *c-MYC* expression (P1), irrespective of the ligand used, which is in agreement with recent literature[20]. Additionally, ATENA has shown that selective MYC-G4 targeting does not result in a net reduction of *c-MYC* expression, as increased transcription from the alternative P2 promoter counterbalances the P1-specific downregulation induced by G4 engagement. The P1-dependent downregulation upon G4-targeting was also confirmed by treatment with the MYC-G4 selective ligand DC-34, further validating the ability of ATENA to target individual G4s. Importantly, we have also observed that guiding G4-ligands in the proximity of the *c-MYC* promoter TATA box can lead to G4-independent transcriptional suppression. This suggests that previous observations obtained with dCas9 decorated with multiple G4-ligands are likely reflecting occupancy of transcriptional regulatory regions rather than genuine G4 binding[50].

Upon employing ATENA to target selectively the iM present in the promoter of *c-MYC*, using a HaloTag-compatible version of Pep-RVS, we have also observed P1-specific transcriptional perturbation. An increase in P1-expression was associated with iM stabilization, opposing the transcriptional inhibition observed with G4 stabilization, in line with other systems, which indicate that iM binding results in the induction of gene expression[51,52] and that iMs and G4 shape opposing effects in cells[33].

While downregulation of P1-mediated *c-MYC* expression was observed with two distinct ligands, PDS and PhenDC3, ATENA also revealed that other G4s could provide different transcriptional outcomes when targeted by the same compounds. This suggested that the biological response elicited by G4 stabilization might reflect more the structural nature of the G4-ligand complex rather than providing direct insights into the endogenous function of G4s, underscoring the importance of the choice of ligand used in different reports aimed at unveiling G4-biology. Finally, we showcased the ability of ATENA to infer the biological relevance of cell line-specific G4s, revealing a strong dependence of the transcriptional perturbation attained upon ligand treatment on the expression level of the targeted gene.

Altogether, our study provides robust evidence supporting ligand and transcriptional-dependent responses to both G4s and iM targeting. ATENA functions as a modular platform to target individual DNA secondary structures in living cells, enabling a precise study of G4 and iM biology. We anticipate that ATENA will offer significant potential for screening cell and ligand-specific responses to DNA secondary structure targeting in a high-throughput manner, which can be further translated for therapeutic design and development.

## Results

### Chemical labeling of CRISPR-Cas9 proteins with G4 ligands
Catalytically inactive CRISPR-Cas9 (dCas9) fused to functional proteins has been widely used in biology to achieve site-selective perturbation of gene expression[53]. This strategy takes advantage of the selectivity provided by dCas9 bound to a short-guiding RNA (sgRNA) in recognizing a specific genomic site by base pairing, which can be used to recruit an effector protein (i.e., a transcription factor or an epigenetic enzyme) at the targeted site[54]. A similar strategy has been recently devised to decorate dCas9 with G4 ligands using non-covalent biotin-streptavidin recognition[50]. Irreversible chemical functionalization of dCas9 proteins has been previously achieved using commercially

available chloroalkane-modified fluorophores to label a dCas9-Halo fusion protein in living cells[54]. Therefore, we hypothesized that generating modified G4 ligands functionalized with chloroalkane moieties could be exploited to decorate with higher control and irreversible covalent chemistry Cas9 proteins with G4 ligands under physiological conditions. To achieve this, we designed analogs of the widely established G4 ligand PDS based on the previously described PyPDS scaffold[5]. Unlike PDS, PyPDS features a single primary amine within its structure, which can be selectively functionalized with chloroalkane side chains by peptide coupling (Fig. 1c)[5]. After successfully synthesizing PyPDS following previously established methods[5], we functionalized the primary amine of the molecule with chloroalkane side chains of different lengths, enabling systematic investigation of the ideal distance between the G4-binding scaffold PyPDS and the HaloTag protein to achieve optimal G4 engagement. Specifically, we have used linkers containing different PEG repeats ($n = 0, 2, 4$) to vary the distance between PyPDS and the chloroalkane (Cl-PyPDS$_n$, Fig. 1c, Supplementary Information 1).

To avoid limiting the use of ATENA to a single G4 ligand, we also functionalized another widely characterized G4 ligand called PhenDC3, using the same chemical strategy (chloroalkane moieties, Cl-PhenDC3$_n$)[47]. Unlike PDS, PhenDC3 has a cationic side chain and is structurally bulkier, displaying a phenanthroline core wider than the pyridine one present in the PDS scaffold (Fig. 1c, d). While PDS and PhenDC3 have been extensively validated as selective G4 ligands, the structural differences between these two ligands may lead to distinct biological responses when targeting G4s in cells, which can be characterized globally with current methods[55,56] but not at the individual G4 site. Therefore, we decided to systematically compare these two ligands with ATENA when recruited to a single G4 site. To achieve this, we have synthesized a previously reported PhenDC3 analog displaying an exocyclic primary amine[57], which can be functionalized by peptide coupling using the same synthetic strategy described for PyPDS to afford chloroalkane-modified PhenDC3 analogs that are compatible with HaloTag conjugation (Fig. 1d, Supplementary Information 1).

### In vitro validation of covalent conjugation of G4-ligands to dCas9-Halo
With both PyPDS and PhenDC3 analogs in hand, we initially assessed whether functionalization with the chloroalkane side chains could affect the G4-binding properties of these molecules. To test this, we subjected all the ligands to FRET or Circular Dichroism (CD) melting to evaluate their ability to stabilize G4 structures after chloroalkane-functionalization. All analogues tested displayed good G4 stabilization, providing an increase in melting temperature ($\Delta T_m > 10\,°C$ at 4 µM) against four distinct G4 structures tested (c-MYC, hTelo, BCL-2, and c-KIT2; Supplementary Tables 9, 10, 11 and 12), with $\Delta T_m$ values comparable to the unfunctionalized ligands. This suggested that the addition of the chloroalkane side chains had a negligible effect on the G4-stabilization properties of both PhenDC3 and PyPDS.

Having confirmed that both chloroalkane-functionalized PyPDS and PhenDC3 analogs retained good G4-binding recognition properties, we next investigated if these molecules could be covalently engaged to dCas9-Halo and performed a competition assay in vitro. To this end, we expressed and purified the dCas9-Halo protein (see methods) and incubated it for 45 min with increasing concentrations of Cl-PyPDS$_n$ and Cl-PhenDC3$_n$. This was followed by incubation with an excess (5 µM) of the commercially available HaloTag®TAMRA (Cl-TAMRA) ligand to label dCas9-Halo with the TAMRA fluorophore. Since labeling of HaloTag is a covalent irreversible process[45], we reasoned that initial exposure of the dCas9-Halo protein to the chloroalkane-functionalized G4 ligands would prevent subsequent incorporation of the fluorescent Cl-TAMRA, leading to a dose-dependent reduction of TAMRA incorporation (Fig. 1e). Indeed, all tested analogs induced a robust dose-dependent decrease of the

dCas9-Halo TAMRA signal (Fig. 1f and Supplementary Fig. 3a), indicating high efficiency in labeling dCas9-Halo irrespective of the G4-ligand (i.e., Cl-PyPDS$_n$ or Cl-PhenDC3$_n$) or the PEG-linker used to connect the chloroalkane to the G4-binding scaffold. To quantify labeling efficiency, we measured fluorescence intensity for each lane of the gel and generated dose-response curves to extract the concentration of ligand required to attain 50% labeling of the dCas9-Halo in vitro (IV-CP$_{50}$). As depicted in Supplementary Fig. 3b, c, all the analogues displayed low IV-CP$_{50}$ values ($\leq$ 2 $\mu$M), demonstrating an ability to functionalize dCas9-Halo in vitro. This data supports the use of chloroalkane-modified G4 ligands to label dCas9-Halo covalently.

## In vitro optimization of sgRNAs and PEG-linker to attain G4-engagement

We next asked whether dCas9-Halo, functionalized with G4 ligands, could be used to target individual G4s. To achieve this, we investigated the ideal distance between the dCas9 binding site and the targeted G4 to achieve optimal engagement of the G4 ligands with the targeted structure by systematically varying both the short-guiding RNA (sgRNA) sequences used and the PEG linker connecting the Halo-reactive moiety (chloroalkane) to the G4-binding scaffold tested. To quantify G4 engagement, we designed a dually fluorescent-labeled DNA template containing an established G4-forming sequence (c-KIT2-G4) at its 3′ end to monitor G4 stabilization through FRET (Fig. 1g). We then designed two sgRNAs to orient the dCas9-Halo complex towards the G4 structure sitting at either 18 (NT-sgRNA$_{FRET-18}$) or 42 (NT-sgRNA$_{FRET-42}$) base pairs from the targeted G4 and whose Protospacer Adjacent Motifs (PAM) are located on the non-template strand bearing the G4 (NT, Fig. 1g). This is based on previous structural studies indicating that the C-terminal domain of the Cas9 protein, where the Halo protein is situated, will point towards the 3′ end of the PAM sequence[58]. Moreover, we have used a scrambled sequence as a non-targeting RNA control (NTC) to rule out any potential unspecific binding that is not strictly mediated by sgRNA-driven proximity. We then used Cl-PyPDS$_n$ molecules as a prototype G4 ligand to assess the extent of G4 targeting by ATENA under different conditions, by measuring changes in FRET when targeting the oligo construct with the dCas9-Halo complex in the presence or absence of Cl-PyPDS$_n$. We failed to detect any significant changes ($p > 0.05$) in the FRET signal for both sgRNAs tested (NT-sgRNA$_{FRET-18}$ vs NT-sgRNA$_{FRET-42}$) when using Cl-PyPDS$_0$ (Supplementary Fig. 3d). Considering that Cl-PyPDS$_0$ can efficiently bind to dCas9-Halo (Fig. 1f, IV-CP$_{50}$ = 1.6 $\mu$M), the lack of G4 stabilization displayed by this molecule indicates that the linker connecting the PDS scaffold to the dCas9-Halo is inadequately short for engaging with the G4 structure. Nevertheless, a trend showing greater changes in FRET efficiency when using sgRNAs closer to the G4 (NT-sgRNA$_{FRET-18}$ vs NT-sgRNA$_{FRET-42}$) suggests that placing the dCas9 closer to the G4 facilitates ligand engagement (Supplementary Fig. 3d). Indeed, when decorating dCas9-Halo with a PDS analog with a longer PEG linker (Cl-PyPDS$_2$), a significant increase ($p < 0.05$) in FRET signal could be measured when using NT-sgRNA$_{FRET-18}$ ($\Delta$FRET = 0.42), which is indicative of G4 engagement. However, when using NT-sgRNA$_{FRET-42}$, we failed to measure an increase in FRET signal, confirming that placing dCas9-PDS closer to the G4 provides better ligand engagement. To further investigate ideal conditions to obtain G4-targeting with ATENA, we also designed T-sgRNA$_{FRET-21}$ and T-sgRNA$_{FRET-41}$ that sit at 21 and 41 base pairs from the G4 but whose PAM is located on the template strand (T, Fig. 1g) to investigate the effect of the dCas9-Halo orientation on G4 targeting (Fig. 1g, h). Consistent with our observations indicating that closer placement of dCas9-Halo to the G4 is linked with better ligand engagement, we detected a significant ($p < 0.05$) increase in FRET signal when using T-sgRNA$_{FRET-21}$ ($\Delta$FRET = 0.33, Fig. 1h) that was abrogated when using T-sgRNA$_{FRET-41}$ (Fig. 1h and Supplementary

Fig. 12a). This further indicated that optimal G4 targeting by ATENA is achieved by using sgRNAs closer to the G4 regardless of the orientation imposed by the sgRNAs used (Fig. 1h and Supplementary Fig. 12a). When further increasing the PEG linker using Cl-PyPDS$_4$, we failed to observe any significant ($p < 0.05$) increase in FRET efficiency with both NT-sgRNA$_{FRET-18}$ and NT-sgRNA$_{FRET-42}$, suggesting that using PEG linkers that are excessively long is detrimental to G4 engagement, possibly due to high entropic penalty associated with ligand recognition (Supplementary Fig. 3e). Altogether, our study demonstrated that dCas9-driven G4-ligand engagement is both PEG-linker and sgRNA-dependent, with optimal G4 engagement achieved when using a PEG2 linker and sgRNAs placing the dCas9-Halo complex as close as possible (depending on the PAM availability) to the targeted G4.

## Chloroalkane-modified ligands can label dCas9-Halo efficiently in cells

Having optimized conditions to achieve G4 engagement in vitro with ATENA−using Cl-PyPDS$_n$ and c-KIT2 G4s as a model system−we next investigated whether ATENA can be used to stabilize individual G4 structures in living cells. The use of cell lines that constitutively express dCas9-Halo is essential to ensure consistent cellular levels of the protein across different experiments, avoiding bias introduced by the significantly variable levels of protein expression typically associated with transient transfection used in previous reports[50]. To this end, we integrated dCas9-Halo into the genome of the breast cancer cell line (MCF7) using standard lentiviral integration approaches (see methods). We selected MCF7 cells in light of the highly diverse transcriptional response previously reported upon treatment with PDS[59], which we wanted to investigate further with ATENA. Successful integration of dCas9-Halo was confirmed by PCR-based genotyping and Western blot (Supplementary Fig. 4a, b).

Next, we evaluated the ability of the chloroalkane-functionalized ligands to bind dCas9-Halo in cells. Using an established chloroalkane penetration assay (CAPA), we compared the relative potency of each ligand to covalently label dCas9-Halo under physiological conditions[60]. During CAPA, cells are initially exposed to increasing concentrations of the chloroalkane-modified G4 ligands, before incubation with a Halo-reactive Oregon Green fluorophore (Cl-OG), which reacts with any HaloTag binding site that has been left unoccupied by the previous exposure to the chloroalkane G4 ligands (Fig. 2a, i-ii). The efficiency of G4-ligand incorporation can therefore be measured as an inverse function of the Oregon Green fluorescence emission, as successful G4-ligand incorporation to Halo prevents subsequent fluorophore functionalization (Fig. 2a, iii-iv). To quantify this numerically, we calculated the half-maximal chloroalkane penetration value (CP$_{50}$), which is the ligand concentration required to label 50% of the available dCas9-Halo molecules and can be used as a direct readout of target occupancy[60]. As displayed in Fig. 2a (right), treatment of MCF7 cells expressing dCas9-Halo with both Cl-PyPDS$_0$ and Cl-PyPDS$_4$ revealed a modest dose-dependent reduction of Oregon Green emission, providing CP$_{50}$ values of 15.9 $\mu$M and 5.4 $\mu$M, respectively (Supplementary Fig. 4c). Conversely, Cl-PyPDS$_2$ could label ~90% dCas9-Halo at a concentration as low as 0.25 $\mu$M (CP$_{50}$ 0.012 $\mu$M, Supplementary Fig. 4c), saturating at 2.5 $\mu$M (Fig. 2a), suggesting that the cellular permeability and bioavailability of Cl-PyPDS$_2$ were particularly suitable for its application in ATENA. Given that the PEG2 linker also led to the best G4 engagement in vitro (Fig. 1h), we decided to assess the compatibility with ATENA of a different G4 ligand (PhenDC3) bearing a PEG2 linker (Cl-PhenDC3$_2$) through CAPA. Gratifyingly, Cl-PhenDC3$_2$ labeled efficiently dCas9-Halo in cells, yielding a CP$_{50}$ value of 1.7 $\mu$M (Supplementary Fig. 4d). Cl-PhenDC3$_4$ showed a similar trend to Cl-PyPDS$_4$, indicating that PEG4-functionalized ligands could not be employed in ATENA.

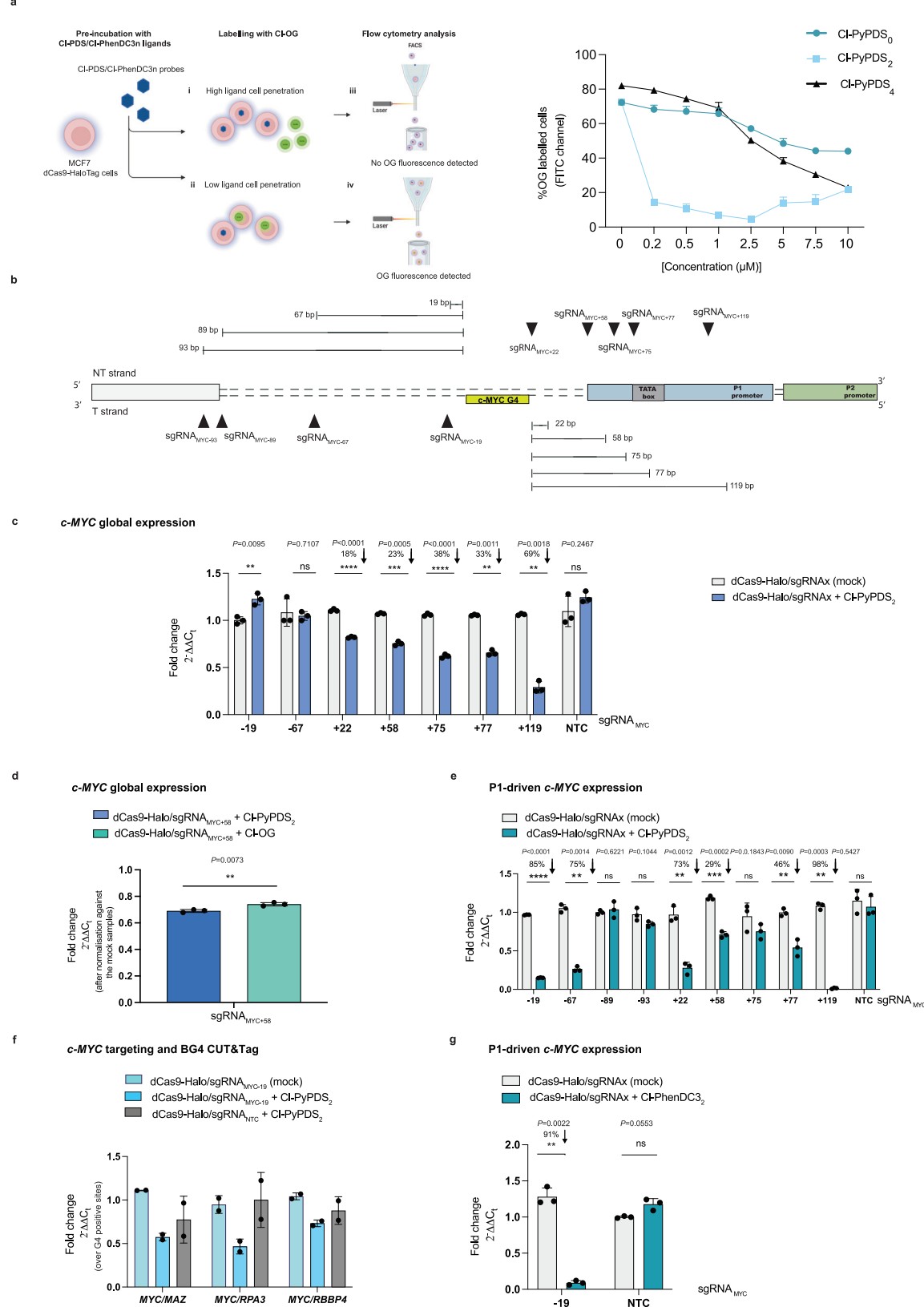

## Selective targeting of MYC-G4 through ATENA reveals no changes in *c-MYC* expression

After identifying conditions to decorate dCas9-Halo with G4 ligands in cells, we next set out to investigate transcriptional responses associated with individual G4 targeting using ATENA. We initiated our study by examining the G4 present in the promoter of the *c-MYC* proto-

oncogene (MYC-G4), as this is one of the well-studied and well-described G4s in the literature. Several studies have linked the targeting of MYC-G4 with ligands to transcriptional suppression of *c-MYC*[12]. To assess the extent of transcriptional perturbation mediated exclusively by G4 stabilization, we designed a panel of sgRNAs to direct ATENA at MYC-G4.

**Fig. 2 | ATENA enables selective targeting of a G4 in the *c-MYC* promoter.**
**a** (**left**) Schematic representation (created with BioRender, https://BioRender.com/
s7u38i3) of the CAPA assay. (**right**) CAPA assay data on MCF7-dCas9-Halo cells
treated with Cl-PyPDS$_n$ and followed by fluorophore incubation (Mean ± SD, $n = 3$).
Data were analyzed with FlowJo software. **b** Schematic illustration of the *c-MYC*
promoter with the annotated G4 (MYC-G4), sgRNA targeting region (black trian-
gles), and their relative distance in bp from the MYC-G4. **c** RT-qPCR for *c-MYC*
expression in MCF7-dCas9-Halo cells transfected with the indicated sgRNAs and
incubated for 48 h in the presence of (2.5 µM) Cl-PyPDS$_2$ or DMSO (mock).
Mean ± SD, $n = 3$, biological replicates, each with three technical replicates. **d** RT-
qPCR for *c-MYC* expression in MCF7 cells stably expressing dCas9-Halo transfected
with sgRNA$_{MYC+58}$ and incubated for 24 h in the presence of either Cl-PyPDS$_2$
(2.5 µM) or Cl-OG (5 µM). Mean ± SD, $n = 3$, biological replicates, each with two
technical replicates. **e** RT-qPCR for P1-driven *c-MYC* expression in MCF7 cells stably
expressing dCas9-Halo transfected with the indicated sgRNAs and incubated for
48 h in the presence of (2.5 µM) Cl-PyPDS$_2$. Mean ± SD, $n = 3$, biological replicates,

each with three technical replicates. **f** BG4 CUT&Tag-qPCR for MCF7 cells stably
expressing dCas9-Halo transfected with either sgRNA$_{MYC-19}$ or sgRNA NTC and
treated with DMSO (mock) or (2.5 µM) Cl-PyPDS$_2$. BG4 accessibility was analyzed for
*c-MYC* and normalized to three G4s in control gene sites (*RPA3*, *MAZ*, *RBBP4*). $n = 2$,
biological replicates, each with three technical replicates for BG4 and one for the
negative (no BG4 treatment). **g** RT-qPCR for P1-dependent *c-MYC* expression in
MCF7 cells stably expressing dCas9-Halo transfected with sgRNA$_{MYC-19}$ or sgRNA
NTC and incubated for 48 h in the presence of (2.5 µM) Cl-PhenDC3$_2$. Mean ± SD,
$n = 3$, biological replicates, each with three technical replicates. The expression
values are represented as fold change ($2^{-\Delta\Delta C_t}$) with respect to the mock (DMSO-
treated) and normalized for the housekeeping gene *GAPDH*; data are the mean of $n$
= number of independent biological samples. Statistical significance was calculated
using a Welch-corrected two-tailed *t* test in GraphPad Prism; *p*-value: ns > 0.05,
*≤0.05, **≤0.01, ***≤0.001, ****≤0.0001. Source data are provided as a Source
Data file.

Specifically, we designed sgRNAs targeting either the non-
template strand (NT) bearing the G4 structure at its 3' end or the
opposite strand at its 5' end (T), Fig. 2b. Based on our biophysical
investigation, we reasoned that placing sgRNAs close enough to the G4
would have ensured G4 stabilization by G4 ligands tethered to dCas9-
Halo (Fig. 1h). Considering PAM sequences available for dCas9-Halo
binding at either G4 ends, we designed sgRNA$_{MYC-19}$ and sgRNA$_{MYC-67}$
that would place the protein complex on the T strand, respectively 19
and 67 base pairs away from the MYC-G4. Similarly, we generated
sgRNA$_{MYC+22}$ and sgRNA$_{MYC+58}$, targeting the MYC-G4 from its 5' end at
a distance of 22 and 58 base pairs on the NT strand, respectively. To
further investigate the optimal distance to achieve G4 stabilization in a
cellular context, which might differ from our simple biophysical
model, we have also designed sgRNA$_{MYC-89}$ and sgRNA$_{MYC-93}$, along
with sgRNA$_{MYC+75}$, sgRNA$_{MYC+77}$, and sgRNA$_{MYC+119}$, also targeting the
MYC-G4 at its 3' and 5' end, respectively, but at a further distance from
the targeted structure (Fig. 2b). After cloning sequences encoding the
various sgRNAs into a vector for mammalian expression (see meth-
ods), we have transfected MCF7 cells stably expressing dCas9-Halo
with individual sgRNAs, followed by treatment with either Cl-PyPDS$_2$ or
mock (DMSO) for 48 h. We then measured changes in *c-MYC* expres-
sion using RT-qPCR, normalizing the expression level against indivi-
dual samples transfected with the respective sgRNAs and mock-
exposed (treated with DMSO). This enabled us to consider any change
in gene expression potentially triggered by the positioning of the
dCas9-Halo complex on the targeted site, and therefore, control for
any transcriptional perturbation imposed by dCas9 that was not
functionalized with ligands.

However, we observed that sgRNAs placing ATENA within
a ~50 bp window of the MYC-G4 on the T strand (sgRNA$_{MYC-19}$,
sgRNA$_{MYC-67}$) in conjunction with Cl-PyPDS$_2$ treatment led to negli-
gible effects on the global expression of *c-MYC* (Fig. 2c), contrasting
our biophysical predictions (Fig. 1h). Conversely, when placing the
complex on the NT strand with sgRNA$_{MYC+22}$ and sgRNA$_{MYC+58}$, we
observed a modest decrease of *c-MYC* expression to 0.8-fold the mock
(~20% reduction), which could be indicative of G4 engagement of PDS
mediated by ATENA (Fig. 2c). To investigate this further, we analyzed
the transcriptional effects elicited by placing the ATENA further away
from the MYC-G4 using sgRNA$_{MYC+75}$, sgRNA$_{MYC+77}$, and sgRNA$_{MYC+119}$,
which should lead to abrogation of G4-engagement in a distance-
dependent fashion, as observed in our biophysical measurements.
Surprisingly, we observed the opposite, with *c-MYC* expression being
reduced the further away the complex was from the G4 (Fig. 2c), which
is inconsistent with a G4-mediated effect. By closer inspection of the
promoter annotation, we noticed that sgRNA$_{MYC+58}$ to sgRNA$_{MYC+119}$
overlapped with the P1 promoter and TATA box sequences. Since
sgRNA$_{MYC+75}$ and sgRNA$_{MYC+77}$ are complementary to the region
identifying a TATA box, we reasoned that ATENA occupies the

TATA box region and hinders transcription initiation, resulting in the
observed *c-MYC* suppression. This effect is exacerbated when using
sgRNA$_{MYC+119}$, which targets the area next to the Transcriptional
Starting Site (TSS) of the P1 promoter (4 bp downstream), further
indicating a G4-independent transcriptional suppression.

Overall, our results indicated that the reduction in *c-MYC* tran-
script levels is driven by ATENA's interference with the TATA-box
region, rather than by ligand-induced stabilization of the MYC-G4,
suggesting that previous observations obtained with the equivalent
of our sgRNA$_{MYC+58}$ are likely affected by this[50]. To test this
hypothesis further, we replaced the G4 stabilizer (Cl-PyPDS$_2$) with a
fluorophore (Cl-OG) and monitored *c-MYC* expression while using
the sgRNA previously reported to provide the strongest down-
regulation (sgRNA$_{MYC+58}$)[50]. Under these conditions, we observed
similar transcriptional downregulation of *c-MYC* compared to that
measured when treating with Cl-PyPDS$_2$ (Fig. 2d), demonstrating
that anchoring the dCas9 complex in the proximity of key tran-
scriptional regions of the *c-MYC* promoter prevents a reliable eva-
luation of G4-mediated transcriptional effects. The interference of
ATENA with *c-MYC* expression when placed in proximity to key
promoter sites is consistent with what has been reported for CRIS-
PRi studies[61,62] and needs to be carefully considered when using
dCas9-based strategies to target G4s[50].

## ATENA confirms P1-dependent transcriptional expression asso-
ciated with MYC-G4

It has been shown that multiple promoters globally contribute to
regulating *c-MYC* expression[63,64]. Therefore, we decided to examine
how G4 targeting affects *c-MYC* expression regulated by specific pro-
moters by analyzing transcripts originating from the two main ones:
P1 and P2.

Indeed, it has recently been shown that genetic deletion of the
MYC-G4 is associated with the selective suppression of transcription
from the P1 promoter, resulting in only a modest reduction in overall
*c-MYC* expression, which is instead represented by the combined
transcriptional output of both the P1 and P2 promoters[20]. When using
ATENA with Cl-PyPDS$_2$ and monitoring P1-driven *c-MYC* expression, we
observed a distance-dependent suppression of P1-mediated expres-
sion with sgRNA$_{MYC-19}$ and sgRNA$_{MYC-67}$. In particular, when using
sgRNA$_{MYC-19}$ we observed an 85% reduction of P1-mediated *c-MYC*
expression (0.15-fold), whereas use of sgRNA$_{MYC-67}$ led to a 75%
reduction of the *c-MYC* expression (0.25-fold), Fig. 2e. Importantly, no
significant changes in expression were detected when using ATENA in
conjunction with sgRNA$_{MYC-89}$ and sgRNA$_{MYC-93}$ that place the Cl-
PyPDS$_2$ excessively distant from the targeted G4, which agrees with a
G4-dependent transcriptional suppression (Fig. 2e). We then asked
why the reduction in P1-driven expression observed with ATENA does
not result in an overall decrease in *c-MYC* transcription. To achieve this,

we also measured changes in transcription originating from the P2 promoter using promoter-specific qPCR primers (see methods). Notably, we observed an increase in P2-selective expression that suggests a compensatory mechanism activated by the cells in response to the G4-induced reduction of P1 transcription (Supplementary Fig. 5a), justifying the absence of statistically significant changes in global *c-MYC* expression detected when using primers amplifying regions common to both P1- and P2-derived transcripts.

Targeting the G4 from its 5' end with sgRNA$_{MYC+22}$ led to a 73% reduction of P1-driven *c-MYC* expression (0.27-fold) Fig. 2e. We also detected P1-dependent transcriptional repression when using sgRNA$_{MYC+58}$ and sgRNA$_{MYC+75}$, which might be affected by G4-independent transcriptional perturbation that we already observed when using these sgRNAs (Fig. 2e). Indeed, P1-mediated *c-MYC* expression was abrogated when the ATENA was directed at sites overlapping close to the P1-TSS with sgRNA$_{MYC+119}$ (Fig. 2e), which is consistent with G4-independent transcriptional inhibition. These observations further confirmed that using ATENA on the 5' end of the MYC-G4 cannot reliably detect changes in gene expression that G4 targeting strictly mediates; careful consideration of the promoter regulatory elements is therefore needed.

Next, we sought to confirm that the observed changes in P1-driven *c-MYC* expression following treatment with sgRNA$_{MYC-19}$ and sgRNA$_{MYC-67}$ result specifically from ATENA-mediated targeting of the MYC-G4, rather than non-specific ligand interactions with other G4 structures. To address the potential for ligand-mediated off-target effects, we monitored the expression of *KRAS*, a gene known to contain a stable G4 structure in its promoter region. As shown in Supplementary Fig. 5b, directing ATENA specifically to the MYC-G4 did not alter *KRAS* expression, supporting the selectivity of ATENA-mediated G4 targeting. In contrast, free PyPDS treatment significantly lowered *KRAS* expression (0.20-fold), leading to an 86% transcriptional suppression (Supplementary Fig. 5c), which validates the ability of ATENA to confer G4-ligand selectivity towards individual G4s, whilst minimizing off-target effects.

### P1-dependent transcriptional inhibition is linked with protein displacement from MYC-G4

We further assessed the ability of ATENA to mediate selective MYC-G4 targeting by investigating perturbation in protein binding at MYC-G4 upon ligand stabilization. It has been proposed that ligands bound to G4s can displace key transcription factors and regulatory proteins, leading to the observed transcription suppression[21,65]. Therefore, we reasoned that if ATENA was correctly positioned to enable ligand-G4 interaction, we should have observed reduced protein accessibility to the G4. To measure this, we used the G4-selective antibody BG4[4] and performed CUT&Tag[7] coupled with qPCR to compare the efficiency of BG4 in enriching for MYC-G4, targeted by ATENA, against 3 independent validated G4 sites (MAZ, RPA3, RBBP4) that should not be affected, as they are not targeted by ATENA. This enabled us to assess the relative protein accessibility at these individual G4 sites under different conditions, as previously described[20]. As displayed in Fig. 2f, when treating cells with Cl-PyPDS$_2$ using sgRNA$_{MYC-19}$, we observed a consistent reduction of BG4 signal that was not detected for the Non-Targeting Control (NTC), irrespective of the reference G4 used. This result suggests that ATENA can be used to guide Cl-PyPDS$_2$ selectively to the MYC-G4 structure, resulting in a decrease in the binding of the BG4 antibody to MYC-G4 due to the binding competition of the ligand, which leads to the displacement of the antibody from the G4. This supports a model in which ligand-mediated G4 stabilization suppresses P1-driven *c-MYC* transcription by outcompeting binding of transcriptional effectors at the MYC-G4[21].

### P1-dependent transcriptional suppression is recapitulated with PhenDC$_3$

To further validate our findings, we used ATENA to deploy a different G4 ligand—PhenDC3[47]—to stabilize the MYC-G4. The PEG2-functionalized

analog, Cl-PhenDC3$_2$, also efficiently labeled dCas9-Halo in cells, as confirmed by CAPA (Supplementary Fig. 4d). When using ATENA with Cl-PhenDC3$_2$ and sgRNA$_{MYC-19}$, we observed a significant inhibition of P1-mediated *c-MYC* transcription (0.09-fold), leading to a 91% reduction of expression (Fig. 2g), which is greater than what was observed with Cl-PyPDS$_2$, likely reflecting the greater G4 stabilization capacity of Cl-PhenDC3$_2$, as indicated by biophysical CD measurements (Supplementary Table 12). This result suggests that ATENA can also be leveraged to compare the relative potency and biological impact of different G4 ligands when deployed at the same genomic target.

Our data indicate that ATENA can successfully target MYC-G4 in a ligand-independent manner, leading to a detectable G4 engagement and corresponding P1-specific *c-MYC* suppression, validating recent findings generated by the genetic deletion of the sequence responsible for the MYC-G4 folding[20].

### The MYC-G4 selective molecule DC-34 validates ATENA

Following the P1-driven *c-MYC* suppression observed upon targeting MYC-G4 with ATENA, we explored whether a similar phenotype could be elicited when using ligands that display some inter-G4 selectivity. To this end, we leveraged the MYC-G4-selective ligand DC-34 (Fig. 3a), which exhibits binding affinity for MYC-G4 with higher selectivity over other G4 structures such as *KRAS* and *c-KIT*[40]. We treated MCF7 cells with increasing concentrations of DC-34 for 48 h before measuring changes in *c-MYC* expression by RT-qPCR. As observed with ATENA, treatment with DC-34 caused negligible dose-dependent changes in global *c-MYC* expression (Fig. 3b), supporting the notion that selective targeting of MYC-G4 does not impact the overall expression of *c-MYC* in MCF7 cells. Conversely, when measuring P1-mediated transcription, DC-34 revealed a dose-dependent suppression that plateaued at 74% reduction when treated with 10 µM of the ligand (0.26-fold), Fig. 3c. This indicated that using an inter-G4-selective ligand for selective MYC-G4 targeting led to observations comparable to those obtained using ATENA, further corroborating the validity of our platform for single G4 targeting.

Next, we assessed the selectivity of DC-34 for MYC-G4 relative to another G4-containing promoter, as we have done for ATENA. Our observations indicated that ATENA did not lead to detectable changes in *KRAS* expression when guided to MYC-G4 (Supplementary Fig. 5b). In contrast, treatment with DC-34 resulted in a dose-dependent reduction in *KRAS* expression (-30%, Fig. 3d), which was less pronounced than what was observed for *c-MYC* under the same conditions, indicating that MYC-G4 is the primary target of DC-34, but residual off-target binding to other G4s may occur at high concentrations. This likely reflects the structural similarity shared among different G4s, which substantially complicates the selective targeting of inter-G4s with small-molecule ligands.

To further confirm that DC-34 downregulates *c-MYC* through direct G4 binding, we also performed BG4 CUT&Tag qPCR upon ligand treatment, which we optimized for ATENA to assess protein occupancy upon treatment. As shown in Fig. 3e, DC-34 treatment significantly reduced the BG4 signal at MYC-G4, which is consistent with the decreased protein accessibility induced by MYC-G4-selective targeting with ATENA. This further supports the use of BG4 CUT&Tag qPCR as an indirect measure of ligand-mediated G4 stabilization associated with transcriptional suppression.

### Transcriptome-wide comparison of ATENA with DC-34

To further assess the inter-G4 selectivity provided by ATENA and DC-34, we analyzed transcriptome-wide gene expression changes using mRNA-Seq. Specifically, we generated mRNA-Seq datasets for MCF7 cells treated with either DC-34 or ATENA (Cl-PyPDS$_2$ in conjunction with sgRNA$_{MYC-19}$) and compared those to transcriptome-wide changes induced by the generic G4 ligand PyPDS. We hypothesized that the inter-G4 selectivity provided to Cl-PyPDS$_2$ by ATENA should be

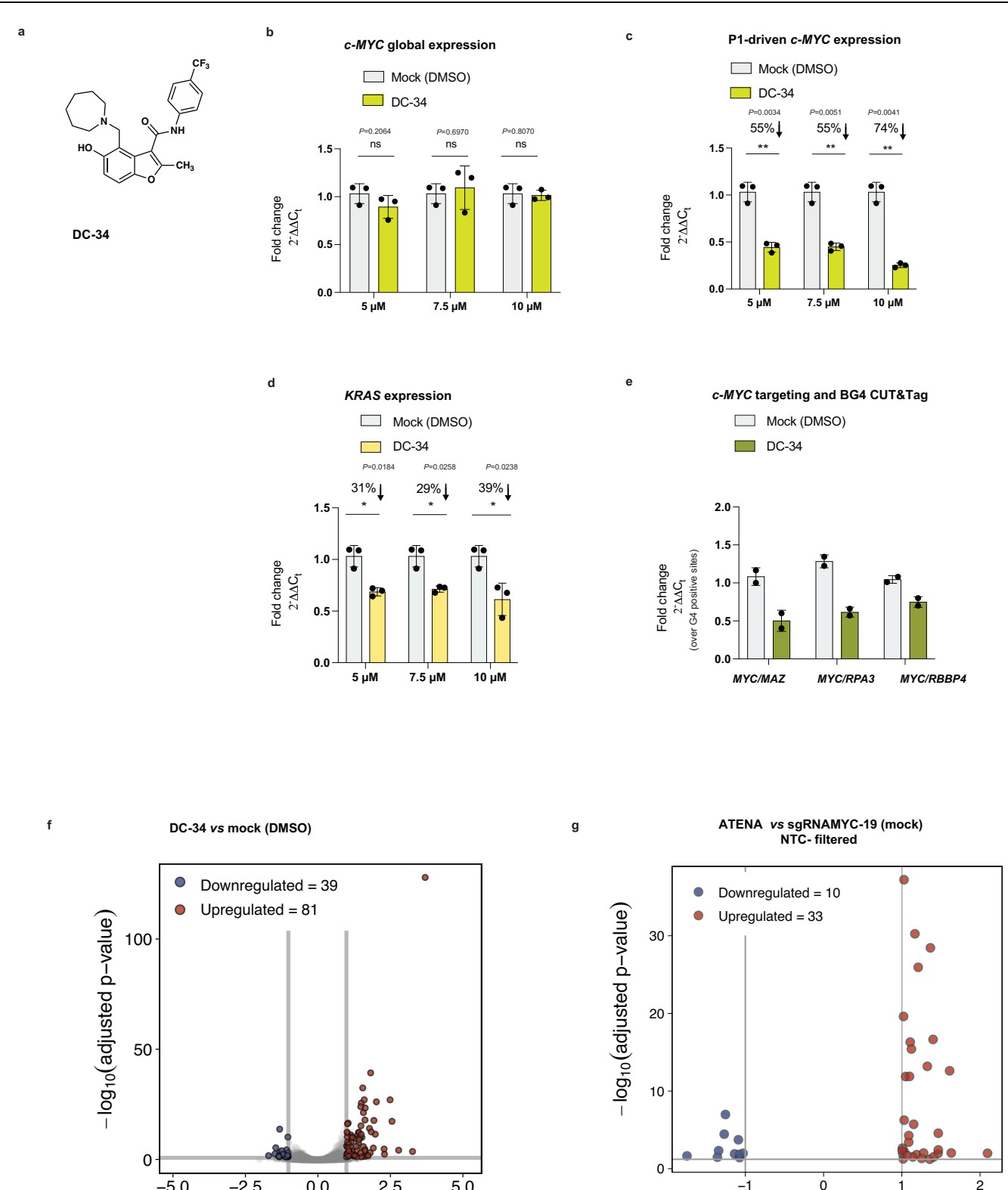

reflected by a substantially lower number of differentially expressed genes (DEGs) when compared to free PyPDS. Indeed, treatment of MCF7 cells with PyPDS (2.5 μM) for 6 h altered the expression of 2228 genes (319 downregulated, 1907 upregulated, FDR < 0.05, Supplementary Fig. 6). In contrast, treatment with the MYC-G4-selective ligand DC-34 (7.5 μM, 6 h) affected only the expression of 120 genes (39 downregulated, 81 upregulated; FDR < 0.05, Fig. 3f), indicating that

the enhanced inter-G4 selectivity of DC-34 is translated into a lower number of genes being differentially expressed when compared to PyPDS. Additionally, mRNA-Seq confirmed that DC-34 did not significantly alter *c-MYC* expression globally, as already observed in our RT-qPCR data, further indicating the selective downregulation of P1 transcripts when targeting MYC-G4, an essential consideration when devising G4-based therapies aiming at suppressing *c-MYC* expression.

**Fig. 3 | The selective MYC-G4 ligand DC-34 matches ATENA. a** Chemical structure of DC-34. **b** RT-qPCR for *c-MYC* expression in MCF7 cells treated with increasing concentrations of DC-34 for 48 h. Values are represented as fold change ($2^{-\triangle\triangle C_t}$) with respect to the mock (DMSO-treated) and normalized for the housekeeping gene *GAPDH*. Mean ± SD, n = 3, biological replicates, each of which included two technical replicates. **c** RT-qPCR for P1-driven *c-MYC* expression in MCF7 cells treated with different concentrations of DC-34 for 48 h. Values are represented as fold change ($2^{-\triangle\triangle C_t}$) with respect to the mock (DMSO-treated) and normalized for the housekeeping gene *GAPDH*. Mean ± SD, *n* = 3, biological replicates, each of which included two technical replicates. **d** Evaluation of potential DC-34 off-targets by analyzing the *KRAS* expression using the same samples reported in (**c**). **e** BG4 CUT&Tag-qPCR for MCF7 cells treated with DMSO (mock) or (7.5 μM) DC-34. BG4 accessibility was analyzed for *c-MYC* and normalized to three G4s in control gene sites (*RPA3*, *MAZ*, *RBBP4*). *n* = 2, biological replicates, including three technical replicates for BG4 and one for the negative (no BG4 treatment). **f** Volcano plot of DEGs in MCF7 cells treated with DC-34 *vs* cells treated with DMSO (mock). Gray = non-significant genes (FDR ≥ 0.05 or |log2FC| <1); red = upregulated DEGs (FDR < 0.05, log2FC ≥ +1); blue = downregulated DEGs (FDR < 0.05, log2FC ≤ −1). **g** Volcano plot of DEGs in MCF7 cells transfected with sgRNA$_{MYC-19}$ treated with Cl-PyPDS$_2$ *vs* mock, after sgRNA NTC filtering; Red = upregulated DEGs (FDR < 0.05, log2FC ≥ +1); blue = downregulated DEGs (FDR < 0.05, log2FC ≤ −1). Plot includes only genes passing FDR < 0.05 and |log2FC| ≥ 1. The data presented are the mean of *n* = number of independent biological samples. Statistical significance was calculated using a Welch-corrected two-tailed *t* test in GraphPad Prism; *p*-value: ns > 0.05, *≤0.05, **≤0.01, ***≤0.001, ****≤0.0001. Source data are provided as a Source Data file.

Given the substantial reduction in DEGs observed with DC-34 relative to a global G4 stabilizer like PyPDS, we hypothesized that conjugation of Cl-PyPDS$_2$ to dCas9 in ATENA should also reduce the extent of transcriptional perturbations. To investigate this, we have transfected MCF7 cells with sgRNA$_{MYC-19}$ and treated them with Cl-PyPDS$_2$ (6 h) before performing mRNA-Seq.

Initially, we performed a standard differential expression analysis comparing ATENA (sgRNA$_{MYC-19}$ or sgRNA NTC + Cl-PyPDS$_2$) to their DMSO-treated controls using DESeq2. We filtered the resulting DEGs for statistical significance (adjusted p-value < 0.05) and magnitude of change (|log2FoldChange| ≥ 1), Supplementary Table 14.

However, to identify genes uniquely responsive to MYC-targeted ligand recruitment, we then excluded all genes that were also differentially expressed in the sgRNA NTC + Cl-PyPDS$_2$ vs DMSO comparison, using the same filtering criteria. This filtering allowed us to isolate transcriptional effects that were specific to the combination of the sgRNA$_{MYC-19}$ and the ligand, rather than shared responses with sgRNA NTC that reflect more unspecific effects of the dCas9 platform (Fig. 3g). This analysis identified only 43 DEGs, underscoring the specificity conferred to Cl-PyPDS$_2$ when conjugated to dCas9 as opposed to the free PyPDS ligand.

To evaluate if ATENA, DC-34, and free PyPDS shared any of the transcriptional changes elicited, we compared DEGs observed in: DC-34 *vs* ATENA (NTC-filtered), PyPDS *vs* ATENA (NTC-filtered), and PyPDS *vs* DC-34, while also testing for enrichment in *c-MYC*-related pathways[66]. When comparing DC-34 *vs* ATENA (NTC-filtered), and free PyPDS *vs* ATENA (NTC-filtered), we found an overlap of a few genes that were not enriched in any pathway, suggesting that these approaches yield largely orthogonal transcriptomic profiles, consistent with their distinct mechanisms of delivery and engagement (Supplementary Tables 15, 16). Finally, when comparing PyPDS vs DC-34, we observed 24 shared DEGs (Supplementary Table 17). However, these DEGs failed to enrich for any known pathways or MYC-related functions, indicating that even structurally distinct G4 ligands with overlapping target preferences can elicit unique transcriptional responses, likely due to differences in binding affinity, cellular uptake, and selectivity.

In summary, these analyses indicated that ATENA, DC-34, and PyPDS can elicit distinct transcriptional responses, with negligible overlap and no shared enrichment for *c-MYC*-related genes. This reinforces the conclusion that ATENA—by spatially confining ligand activity to a single G4 at the *c-MYC* promoter—induces highly selective gene expression changes, contrasting with the broader, less discriminating effects of freely diffusing G4 ligands. Additionally, our mRNA-Seq analysis further confirmed that MYC-G4 targeting is not associated with significant *c-MYC* downregulation in MCF7 cells, and caution is needed when designing MYC-based therapeutics based on G4 targeting.

## Targeting of the *c-MYC* i-motif with ATENA is associated with transcriptional stimulation

After validating the suitability of ATENA for the selective targeting of individual G4 structures in the genome—exemplified by the MYC-G4—

we next explored whether this platform could be adapted to interrogate other DNA secondary structures, such as i-motifs (iMs). Similar to G4s, iMs are stabilized by Hoogsteen hydrogen bonding[22]. However, they form in cytosine-rich regions of the genome, typically complementary to G-rich G4-forming sequences[67]. The formation of iMs in cells has been recently validated in living cells by both immunofluorescence[31] and genome-wide mapping[32,68]. Like G4s, iMs have been implicated in transcriptional regulation[51], although their mechanistic roles remain less well characterized compared to G4s.

To assess the potential of individual iMs to modulate gene expression, we decided to use ATENA in combination with selective iM ligands. Conveniently, the *c-MYC* promoter also bears an iM structure complementary to the G-rich sequence forming the MYC-G4[69], which has been reported to modulate *c-MYC* expression when targeted with selective iM ligands[26,52,70–73]. We therefore reasoned that the same sgRNAs optimized for selective targeting of MYC-G4 could be repurposed to localize ATENA to the MYC-iM, enabling selective iM targeting when decorated with an appropriate ligand. To this end, we utilized a recently developed class of short peptides from the Waller group, which show high affinity and specificity for iMs over G4s[49]. Among these, we selected the RVS peptide (pep-RVS) for its synthetic ease and potential amenability to modification. To enable compatibility with ATENA, we chemically modified pep-RVS with chloroalkane side chains to ensure covalent attachment onto the HaloTag (Supplementary Information 2 and Supplementary Information 7), generating Cl-pep-RVS$_n$ analogs with different PEG linkers (*n*). We initially confirmed that functionalization with chloroalkane did not affect the ability of pep-RVS to bind iMs via UV titrations, confirming that it bound the *c-MYC* i-motif structure with a $K_d$ of 0.35 ± 0.12 μM compared to >33 μM for G4 (Supplementary Fig. 26, Supplementary Fig. 8, and Supplementary Table 13. CD melting experiments also indicated that the *c-MYC* iM structure has two melting temperatures, one main one at 33 °C and another smaller population at 83 °C (Supplementary Fig. 8c). Melting in the presence of Pep-RVS gives rise to only one population, with a melting temperature of 39 °C, indicating stabilization of the main MYC-iM population with a $\Delta T_m$ of +6 °C (Supplementary Fig. 8c). We next assessed the ability of Cl-pep-RVS$_n$ analogs to covalently bind to the dCas9-Halo in living cells using CAPA. As shown in Supplementary Fig. 7a, all Cl-pep-RVS$_n$ analogs displayed good cellular permeability and effectively labeled dCas9-Halo at low μM concentrations. However, Cl-pep-RVS$_4$ performed best in CAPA, providing a CP$_{50}$ value of 2.3 μM, and was selected for further application in ATENA.

To evaluate whether targeting MYC-iM affects *c-MYC* transcription, we employed ATENA with sgRNA$_{MYC-19}$ and treated the cells with Cl-pep-RVS$_4$ for 48 h, using the same conditions previously optimized for G4 targeting. Notably, ATENA-mediated iM targeting led to a significant increase in P1-driven (~2-fold) *c-MYC* transcription (Fig. 4b), in contrast to the transcriptional repression observed upon MYC-G4 targeting. Interestingly, this P1-mediated upregulation was accompanied by a decrease in P2-driven transcription (Fig. 4b), resulting in minimal net changes in global *c-MYC* expression—a functional effect

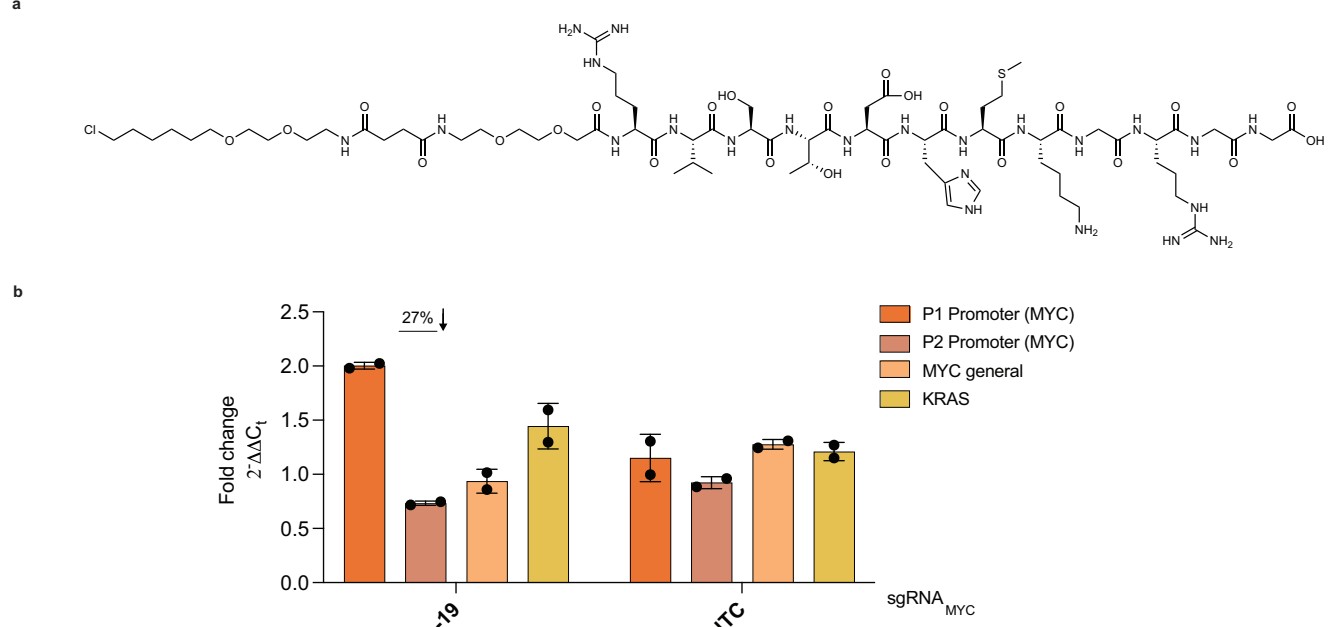

**Fig. 4 | ATENA and *c-MYC* iM targeting. a** Chemical structure of chloroalkane-modified pep-RVS with a 4-PEG linker (Cl-pep-RVS$_4$). **b** RT-qPCR of the indicated genes in MCF7 cells stably expressing dCas9-Halo transfected with sgRNA$_{MYC-19}$ or sgRNA NTC and incubated for 48 h in the presence of 10 μM of Cl-pep-RVS$_4$ or DMSO (mock). The expression values are represented as fold change ($2^{-\Delta\Delta C_t}$) with respect to the mock (DMSO-treated) transfected samples and after normalization for the housekeeping gene (*GAPDH*). *n* = 2, biological replicates, each with three technical replicates. The data presented are the mean of n = number of independent biological samples. Statistical significance was calculated using a Welch-corrected two-tailed t-test in GraphPad Prism; p-value: ns > 0.05, *≤0.05, **≤0.01, *** ≤0.001, ****≤0.0001. Source data are provided as a Source Data file.

opposite to that seen with G4 engagement. This is in line with other examples in the literature where stabilization of iMs results in transcriptional activation[51,74,75], and generally gives rise to the opposite effects of stabilizing G4s[33].

Overall, these results underscore the modularity of ATENA in targeting distinct DNA secondary structures within the same genomic locus by simply varying the conjugated ligand. By exploiting the high iM selectivity of pep-RVS, we demonstrate that selective iM targeting is associated with P1-dependent transcriptional activation of *c-MYC*—a functional effect opposite to that observed with G4 engagement. This highlights the capacity of ATENA to disentangle the complex regulatory roles of overlapping secondary structures in gene promoters.

## PDS can act as a molecular glue of specific G4-protein interactions

After establishing the reliability of ATENA in accurately measuring biological responses mediated by individual DNA secondary structures, using *c-MYC* as a case study, we next questioned whether this platform could be expanded to study transcriptional responses uniquely associated with specific ligands. For instance, previous reports indicated that PDS treatment of MCF7 cells led to significant upregulation, rather than repression, of the long non-coding RNA *PVT1*- an observation we confirmed also for PyPDS by RT-qPCR and mRNA-Seq (Supplementary Fig. 5c, 6)[59]. Therefore, we sought to leverage ATENA to determine if PDS-mediated *PVT1* upregulation was a direct response to the specific stabilization of the G4 in its promoter or rather an indirect effect caused by global G4 stabilization. Previous CUT&Tag experiments performed in MCF7 cells identified a clear G4 peak in the *PVT1* promoter[76], which we used to design sgRNAs for ATENA-based targeting of the PVT1-G4. Considering available PAM sequences, we designed two sgRNAs targeting the PVT1-G4 at either 20 base pairs from its 5' end (sgRNA$_{PVT1-20}$) or 33 base pairs from its 3' end (sgRNA$_{PVT1+33}$) and that display no overlap with any annotated regulatory site (Fig. 5a). Upon transfection with either sgRNA$_{PVT1-20}$ or

sgRNA$_{PVT1+33}$ and treatment with Cl-PyPDS$_2$, we observed a boosted *PVT1* expression of ~4-fold, consistent with observations reported using free PDS (Fig. 5b). G4 ligands typically compete with regulatory proteins for G4 binding, which leads to transcriptional suppression, as we recapitulated measuring BG4 occupancy upon MYC-G4 targeting (Fig. 2f). Therefore, we questioned whether PDS could instead act as a molecular glue when binding to the PVT1-G4, leading to enhanced protein binding to the PVT1-G4 and, thus, justifying the observed transcriptional increase upon PDS treatment. We performed BG4 CUT&Tag qPCR on the PVT1-G4 targeted by ATENA and in conjunction with Cl-PyPDS$_2$ and observed a rise in BG4 occupancy at *PVT1* of ~2-fold (Fig. 5c). This contrasts with the transcriptional response elicited by the same ligand when targeting a different G4 (MYC-G4, Fig. 2f) and points to the role of PyPDS as a molecular glue for protein-G4 interactions within the PVT1-G4. Moreover, this observation suggests that the previously described increase in *PVT1* expression elicited by PyPDS treatment in MCF7 cells reflects the genuine response of the ligand targeting the PVT1-G4 and cannot be ascribed to a secondary response associated with global G4 targeting.

We next asked whether the increase in *PVT1* expression measured with PyPDS was limited to this molecule or if a more general response could be observed with any G4-ligand. MCF7 treatment with free PhenDC3 led to the suppression rather than the enhancement of *PVT1* expression, suggesting that different G4-ligands might elicit different responses when targeting an identical G4 (Supplementary Fig. 7b).

To further investigate this, we selectively targeted the PVT1-G4 using ATENA in conjunction with Cl-PhenDC3$_2$ and either sgRNA$_{PVT1-20}$ or sgRNA$_{PVT1+33}$. Under these conditions, we observed a suppression of *PVT1* transcription, 61% and 34%, respectively, (expression level of the sample was 0.39-fold and 0.66-fold the mock respectively), (Fig. 5d), which is consistent with free PhenDC3 treatment (Supplementary Fig. 7b) and opposite to what was observed with PyPDS (Supplementary Fig. 7c). This suggested that G4-ligands can elicit a different response when bound to the same G4, possibly

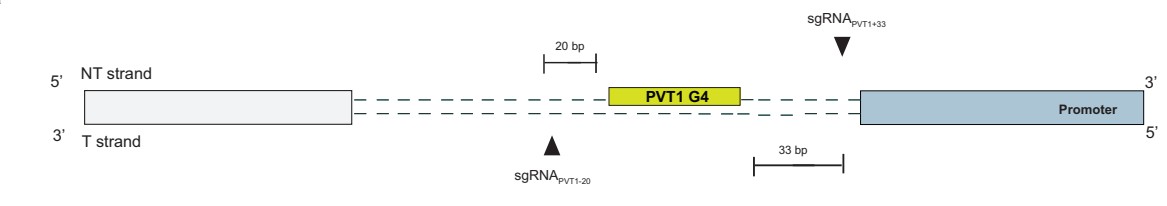

**a**

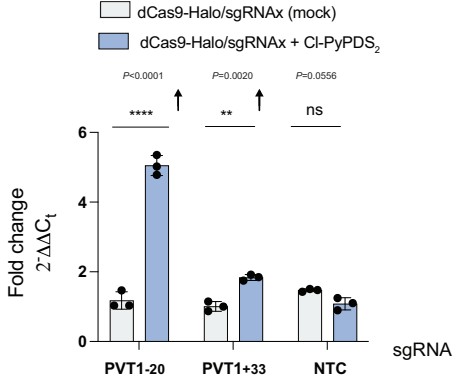

**b** *PVT1* expression

**c** *PVT1* targeting and BG4 CUT&Tag

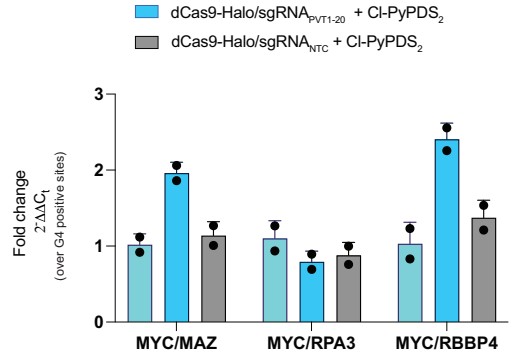

**d** *PVT1* expression

**e** *PVT1* expression

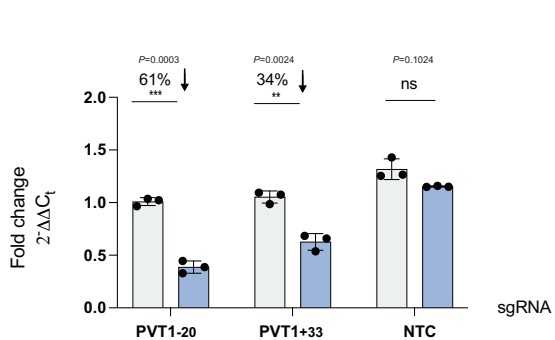

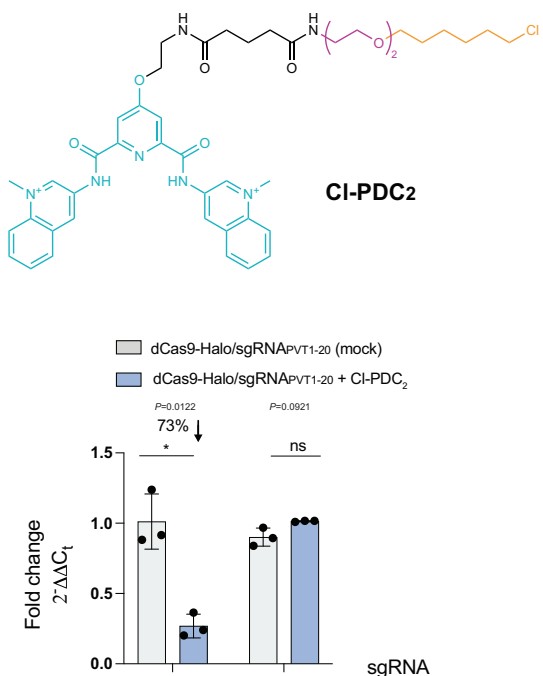

reflecting binding modalities that can either increase or prevent protein accessibility, which can be indirectly measured by BG4 occupancy (Supplementary Fig. 7d). This is an essential factor to consider when using different G4-ligands to infer the biology associated with these secondary structures, as it is often assumed that G4 ligands will all behave the same.

To further investigate these ligand-specific observations, we have also synthesized a chloroalkane analog of a third widely used G4-ligand: Pyridodicarboxamide (PDC, Fig. 5e)[77]. We functionalized the PDC scaffold with a PEG2 chloroalkane side chain (Cl-PDC₂) to mimic the PyPDS and PhenDC3 analogs used in ATENA (Fig. 4e, Supplementary Information 1). After validating the labeling efficiency of Cl-PDC₂

**Fig. 5 | ATENA unveils a ligand-dependent transcriptional response of the lncRNA *PVT1*. a** Schematic overview of the *PVT1* promoter containing annotation of the predicted G4, sgRNA targeting region (black triangles), and their relative distance in bp from the G4-forming sequence **b** RT-qPCR for *PVT1* expression in MCF7 cells stably expressing dCas9-Halo, transfected with either sgRNA$_{PVT1-20}$, sgRNA$_{PVT1+33}$ or sgRNA NTC, and treated with (2.5 µM) Cl-PyPDS$_2$ for 48 h after transfection. The expression values are represented as fold change ($2^{-\triangle\triangle C_t}$) with respect to the mock (DMSO-treated) and normalized for the housekeeping gene *GAPDH*; Mean ± SD, $n = 3$, biological replicates, each of which included three technical replicates. **c**, BG4 CUT&Tag-qPCR for MCF7 cells stably expressing dCas9-Halo transfected with either sgRNA$_{PVT1-20}$ or sgRNA NTC and treated with DMSO (mock) or (2.5 µM) Cl-PyPDS$_2$. BG4 accessibility was analyzed for *PVT1* and normalized to three G4s in control gene sites (*RPA3, MAZ, RBBP4*). $n = 2$, biological replicates, including three technical replicates for BG4 and one for the negative (no BG4 treatment). **d** RT-qPCR for *PVT1* expression in MCF7 cells stably expressing dCas9-Halo, transfected with either sgRNA$_{PVT1-20}$, sgRNA$_{PVT1+33}$ or sgRNA NTC and treated with (2.5 µM) Cl-PhenDC3$_2$ for 48 h after transfection. Values are represented as fold change ($2^{-\triangle\triangle C_t}$) with respect to the mock (DMSO-treated) and normalized for the housekeeping gene *GAPDH*. Mean ± SD, $n = 3$, biological replicates, each of which includes three technical replicates. **e** (**Top**) Chemical structure of chloroalkane-modified PDC (Cl-PDC$_2$) with the PEG linker length of two; (**bottom**) RT-qPCR for *PVT1* expression in MCF7 cells stably expressing dCas9-Halo, transfected with either sgRNA$_{PVT1-20}$ or sgRNA NTC and treated with (2.5 µM) Cl-PDC$_2$ for 48 h after transfection. Values are represented as fold change ($2^{-\triangle\triangle C_t}$) with respect to the mock (DMSO-treated) and normalized for the housekeeping gene *GAPDH*. Mean ± SD, n = 3, biological replicates, each of which includes three technical replicates. Statistical significance was calculated using a Welch-corrected two-tailed t-test in GraphPad Prism; *p*-value: ns > 0.05, *≤0.05, **≤0.01, ***≤0.001, ****≤0.0001. Source data are provided as a Source Data file.

through CAPA (Supplementary Fig. 13), we used it in conjunction with sgRNA$_{-20}$ to target the *PVT1* promoter and investigate associated transcriptional responses. Similarly to what was observed with PhenDC3, PDC lowered *PVT1* expression (0.27-fold), leading to a 73% reduction (Fig. 5e), suggesting that PDC interacts with the PVT1-G4 in a manner reminiscent of PhenDC3 and causes protein displacement from the G4. Structurally, the PDC scaffold is indeed similar to PhenDC3, displaying methylated nitrogens on the quinolines that are facing opposite orientation compared to PDS (Fig. 5e). Moreover, both PhenDC3 and PDC lack the amino-side chains present in PyPDS, further highlighting the structural similarity between these two scaffolds, which might recapitulate the similar response observed. Collectively, our results indicated that the transcriptional responses elicited by ligands at individual G4s depend heavily on the structural nature of the ligand and its binding modality, which may lead to protein displacement at the G4 site or act as a molecular glue enhancing G4-protein interactions. This suggests that transcriptional changes observed upon G4-ligand treatment should be interpreted as ligand-specific outcomes, reflecting the response to a specific ligand at a specific G4 structure rather than the endogenous function of the DNA structure.

## Targeting cell-specific G4s with ATENA reveals transcription-dependent response to ligands

CUT&Tag and other chromatin-compatible G4-mapping methods, such as BG4-ChIP and Chem-Map, have shown that the genomic distribution of G4s is cell-specific and predominantly located at promoters of highly expressed genes[7,9,10,78]. Therefore, we decided to investigate biological responses attained when directing a ligand towards previously unexplored MCF7-specific G4s. Specifically, we aimed to determine whether the biological relevance of individual G4s and their response to ligand binding correlate with the expression levels of the associated genes. We leveraged the existing dataset on G4 distribution in MCF7 cells previously obtained using CUT&Tag[76]. This dataset identified a G4 peak in the promoter of the highly expressed *HMGN1* gene - encoding for a non-histone chromosomal protein able to interact with nucleosomes and regulate chromatin structure[79,80] - as unique to MCF7 cells compared to other cell lines[76]. We designed sgRNA$_{HMGN1-22}$ and sgRNA$_{HMGN1+34}$ to target the HMGN1-G4 at 22 and 34 base pairs, respectively, at its 3' and 5' ends, ensuring no overlap with known functional regions (Fig. 6a). After transfecting MCF7 cells expressing dCas9-Halo with sgRNA$_{HMGN1-22}$ and sgRNA$_{HMGN1+34}$, we incubated them with Cl-PyPDS$_2$ for 48 h, as per the optimized ATENA protocol. Under these conditions, we measured a 99% reduction of *HMGN1* expression (0.01-fold) when targeting its G4 at the closest distance of 22 base pairs with sgRNA$_{HMGN-22}$ (Fig. 6b). *HMGN1* downregulation was partially attenuated when placing ATENA further away from the G4 with sgRNA$_{HMGN+34}$, consistent with the distance-dependent ligand engagement observed for other G4s (Fig. 6b). To

place the observed *HMGN1* downregulation in a biologically meaningful context, we examined the role of this gene in breast-cancer dormancy − an epigenetic-driven, non-replicative state, from which cancer cells "awaken", causing cancer relapse and resistance to therapy. In dormant MCF7 cells (estrogen-deprived), we inspected the epigenetic changes, chromatin accessibility, and transcriptional profile at the *HMGN1* promoter. Dormant cells exhibited loss of the active histone mark H3K4me3, gain of the repressive histone mark H3K27me3, and a corresponding drop in *HMGN1* expression; these epigenetic and transcriptomic changes were partially reversed upon cell-cycle re-entry − "awakening"[81] (Supplementary Fig. 10a, b). Remarkably, ATENA-mediated stabilization of the HMGN1-G4 reproduced this repressive transcriptional state (Fig. 6b). Therefore, these convergent observations suggest that the HMGN1-G4 could act as an epigenetic switch: stabilizing the structure could indeed reinforce the *HMGN1* repressive state characteristic of dormancy and might be investigated as a strategy to maintain residual tumor cells in a dormant state.

Altogether, these findings indicated that targeting a cell-specific G4 located in the promoter of a highly transcribed gene can suppress gene expression, suggesting that maintaining G4-homeostasis at the promoter of highly transcribed genes is key to sustaining elevated expression levels and making these G4s particularly sensitive to ligands. Therefore, the varied transcriptional response observed upon G4-ligand treatment might reflect the relative relevance of different G4s in sustaining gene expression in the specific cell line studied.

To corroborate this hypothesis, we utilized ATENA to target a G4 present in the promoter region of a gene expressed at low levels in MCF7 cells: *IL17RA*. Indeed, BG4 CUT&Tag performed in MCF7 cells revealed a distinct G4-peak in the promoter region of *IL17RA*[76], a gene that is only marginally expressed in this cell line. *IL17RA* encodes for the Interleukin 17 Receptor A, a proinflammatory cytokine secreted by activated T-lymphocytes and, therefore, not essential for breast cancer cell homeostasis. We generated sgRNA$_{IL17RA-20}$ to target the IL17RA-G4 at 20 base pairs from its 5' end and within a region that does not overlap with other regulatory elements of this promoter. Following transfection with sgRNA$_{IL17RA-20}$ or sgRNA$_{NTC}$ and incubation with Cl-PyPDS$_2$, we failed to detect any measurable changes in *IL17RA* expression levels (Fig. 6c). Extending *IL17RA* targeting to other G4 ligands (i.e., Cl-PhenDC3$_2$ and Cl-PDC$_2$) and an additional sgRNA (sgRNA$_{IL17RA+36}$) also failed to elicit any detectable changes in expression (Supplementary Fig. 9). Altogether, these observations indicate that targeting a G4 in a promoter of a transcriptionally inactive gene is not associated with gene-expression perturbation, linking the functional relevance of G4s to the transcriptional levels of the genes associated. Considering that both G4s in the *HMGN1* and *IL17RA* promoters are equally detected in MCF7 by CUT&Tag[76] and targeted with similar sgRNA designs (within PAM sequence constraints), our findings suggest that the transcriptional levels linked to

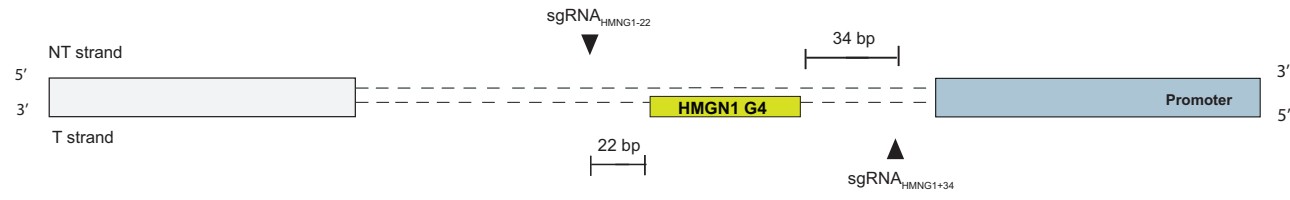

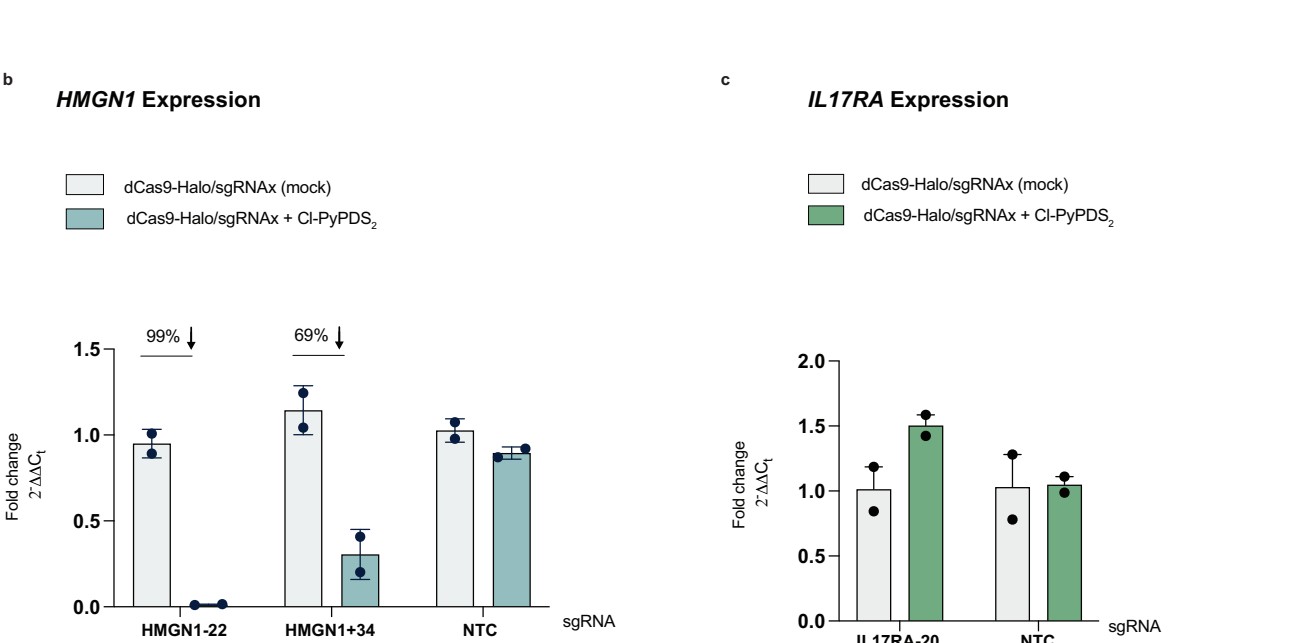

**Fig. 6 | Targeting of de novo G4s with ATENA uncovers a transcriptionally dependent functional response. a** A schematic overview of the *HMGN1* promoter, including annotation of the predicted G4, two sgRNA designed to target HMGN1-G4 (black triangles), and their relative distance in bp from the G4-forming sequence. **b** RT-qPCR for *HMGN1* expression in MCF7 cells stably expressing dCas9-Halo, transfected with either sgRNA$_{HMGN1-22}$, sgRNA$_{HMGN1+34}$ or sgRNA NTC and treated with (2.5 μM) Cl-PyDS$_2$ for 48 h after transfection. The expression values are represented as fold change ($2^{-\triangle\triangle C_t}$) with respect to the mock (DMSO-treated) and normalized for the housekeeping gene *GAPDH*; $n$ = 2, biological replicates, each of which includes three technical replicates. **c** RT-qPCR for *IL17RA* expression in MCF7

cells stably expressing dCas9-Halo, transfected with either sgRNA$_{IL17RA-20}$ or sgRNA NTC and treated with (2.5 μM) Cl-PyDS$_2$ for 48 h after transfection. The expression values are represented as fold change ($2^{-\triangle\triangle C_t}$) with respect to the mock (DMSO-treated) and normalized for the housekeeping gene *GAPDH*; $n$ = 2, biological replicates, each of which includes three technical replicates. Data presented are the mean of n = number of independent biological samples. Statistical significance was calculated using a Welch-corrected two-tailed t-test in GraphPad Prism; *p*-value: ns > 0.05, *≤0.05, **≤0.01, ***≤0.001, ****≤0.0001. Source data are provided as a Source Data file.

the targeted G4 can be used to anticipate the extent of transcriptional perturbation associated with ligand treatment. This model also explains the relatively modest transcriptional changes observed by the selective targeting of *c-MYC* and *PVT1* G4s, which are only moderately expressed in MCF7 cells. Increasing evidence in the literature suggests a model in which G4s act as epigenetic factors to mark highly transcribed genes[12]. Our data support this model by showcasing how the extent of gene suppression/activation elicited by G4 ligands is linked to basal transcriptional levels, underscoring the relevance of maintaining G4 homeostasis in preserving transcriptional profiles characteristic of specific cell lines.

## Discussion

There is now substantial evidence to support that G4 structures form within endogenous chromatin and that their formation is intimately linked to transcriptional activity[12]. For instance, G4s have been detected in the promoter regions of key oncogenes in cancer cells, and recent studies have demonstrated that they constitute critical structural features required to sustain high transcriptional rates, such as in the case of c-*MYC*[20]. Over the past few decades, the development of

selective G4 ligands has revealed that stabilizing these structures often results in the suppression of oncogene expression[13]. Consequently, G4s have represented an attractive therapeutic target for decades. However, the use of G4 ligands for clinical applications has not yet gained traction, reflecting two intrinsic limitations. Firstly, the recognition mechanism leveraged by most G4 ligands relies on end-stacking interactions, which lack the selectivity to discriminate among different G4s—while being effective at distinguishing G4s from duplex DNA. Given that the prevalence of G4s in the genome is highly cell-type specific, the lack of inter-G4 selectivity displayed by G4 ligands results in widespread transcriptional perturbations and inconsistent phenotypes across various cell models. Secondly, the binding affinity of individual ligands to different G4s varies broadly, making it difficult to pinpoint which G4s are functionally responsible for the biological responses elicited by any given ligand.

To address these constraints, tools enabling single-G4 targeting have been considered essential to unravel the fundamental biology regulated by specific G4s and to validate their therapeutic potential.

Although several locus-directed and ligand-based strategies for single-G4 targeting have been reported[50,82–85], these approaches

present significant methodological limitations (see Supplementary Table 18) that hinder their broader application and their suitability to investigate G4 biology with resolution and at scale. Recently, Qin et al.[50]. proposed a similar CRISPR-Cas9-based approach to target individual G4s. While this method also offers sequence specificity, it lacks the chemical versatility required for systematic ligand comparison studies and, more importantly, relies on the use of an array of 10 ligands per G4, which may result in crowding effects at target regions that we have described. Additionally, this approach lacks a quantitative assay that enables precise dosing of ligand concentrations (the equivalent of CAPA for ATENA), thereby increasing the risk of off-target effects due to uncontrolled free ligand distribution. Finally, the work by Qin et al. does not take into account the transcriptional contributions of the P1-promoter and the effects of sgRNA positioning when key regulatory regions are targeted. Aspects that should be carefully considered when studying G4-mediated transcription to avoid a misleading interpretation of the data collected[50].

The significant limitations associated with preexisting methodologies motivated us to develop ATENA —a CRISPR-guided platform in which catalytically inactive dCas9 is chemically functionalized with G4 ligands. This system allows the positioning of a ligand in proximity to a specific G4 of interest using a short-guiding RNA. We demonstrated that this approach enables transcriptional modulation attributable to G4 engagement at a single genomic locus.

We initially optimized conditions for single-G4 targeting in vitro, before applying ATENA in cells to investigate the transcriptional role of a G4-structure located in the promoter region of the proto-oncogene c-MYC. While independent studies have previously reported c-MYC suppression upon treatment with G4-ligands[70], it remains unclear whether this effect is exclusively mediated by the engagement of ligands with the MYC-G4 or from broader transcriptomic changes induced by global G4 stabilisation. Indeed, Hurley and co-workers—who initially proposed that c-MYC downregulation was exclusively attributed to the targeting of MYC-G4[70]—challenged the previously proposed model, suggesting that transcriptional suppression was more likely a response to global G4 stabilization[18]. Using ATENA, we observed that G4-mediated c-MYC suppression in MCF7 cells is minimal and associated with P1-controlled transcription. These findings are not only in line with what has been shown by recent genomic studies[20] but were also fully recapitulated when using the MYC-G4 selective small molecule DC-34[40]. This indicates that the biological response obtained by selective MYC-G4 targeting is limited to P1-mediated transcription, regardless of the targeting approach.

Importantly, we have noted that the MYC-G4 lies in proximity to key regulatory regions of the promoter, including the P1 promoter and the TATA box sequence. Therefore, using CRISPR-based tools for selective MYC-G4 targeting can easily lead to misleading results when using sgRNAs targeting those regions and placing small molecules near these key regulatory elements. For instance, a recent study reported global c-MYC downregulation when using either dCas9-Nucleolin fusion or dCas9 poly-labeled with ten G4-ligands in tandem[50]. However, this effect was observed when targeting the same region as our sgRNA$_{MYC+58}$, which places the dCas9 protein 9-bp apart from the TATA box and we demonstrated it causes unspecific c-MYC downregulation (Fig. 2d). These findings indicated that placing ligands near core promoter elements can lead to transcriptional changes unrelated to G4 stabilization and that careful design of sgRNAs is required when using dCas9-based tools to avoid false positives. Additionally, our data suggest that the use of multiple ligands on a single dCas9 protein to induce transcriptional perturbation may lead to false positives by inducing local overcrowding at promoters[50]. Similarly, using multiple sgRNAs on a single target can perturb the homeostasis of regulatory elements in a ligand-dependent manner, without necessarily reflecting G4-specific effects, and should thus be avoided[50].

Using the selective G4-antibody BG4, we confirmed that the P1-specific c-MYC downregulation induced by ATENA is accompanied by reduced protein accessibility at the MYC-G4 site, supporting the notion that G4 engagement underlies the observed transcriptional suppression by hampering G4-protein interactions. Notably, a similar effect was observed with DC-34, further suggesting that reduced BG4 binding is to be expected when the ligand successfully engages the G4 within the c-MYC promoter, instead of the previously reported enhancement[50]. This also provides evidence that selective G4 targeting at the c-MYC promoter can be leveraged therapeutically to suppress its expression by interfering with protein-G4 interactions. However, this effect is limited to c-MYC transcription mediated by the P1 promoter and, therefore, dependent on the cellular system investigated[40]. For example, the substantial c-MYC downregulation observed upon treatment with DC-34 in multiple myeloma cells[40] is consistent with the high levels of P1-driven c-MYC expression characteristic of these cancers[86,87], whereas it is ineffective in other cells that are less reliant on the P1 promoter for c-MYC expression (i.e., MCF7).

ATENA further enabled us to investigate the transcriptional response elicited when targeting the iM structure, also present within the c-MYC promoter. By decorating ATENA with an iM-selective peptide recently developed by the Waller's group (RSV)[49], we also observed transcriptional changes limited to the P1-promoter. However, when targeting the iM, we measured an increase in P1-driven transcription, rather than a reduction, suggesting that distinct DNA secondary structures may differently affect expression at the same promoter, as previously postulated in the relevant literature[33]. On the other hand, the observed response may also be related to perturbation of the MYC-G4 dynamics, representing a fascinating interplay that warrants further investigation, potentially through coupling ATENA with genomic approaches. Nevertheless, targeting of iMs demonstrated the modularity of ATENA, which can be easily adapted to target any DNA structure of interest using the same design principles.

ATENA also enabled us to explore ligand-dependent variation in responses when targeting the same G4 structure. It is well established that G4 ligands can cause either transcriptional activation or repression depending on the G4-associated promoter, and this variability has often been attributed to the context-dependent biological roles of the specific G4 being investigated. However, this model contradicts genomic studies that indicate a global association of G4 formation with transcriptional activation rather than a context-dependent function of these structures. Using ATENA in combination with CUT&Tag, we could demonstrate that the variation in gene expression responses at specific G4 sites stems not from inherent differences in G4 function, but rather from how structurally different ligands affect protein-G4 interactions.

We tested this by characterizing the different transcriptional responses observed upon targeting the G4 in the promoter of the long non-coding RNA PVT1 when using two established G4 ligands: PDS and PhenDC3. The previously observed upregulation in PVT1 transcription following PDS treatment[59] was recapitulated with ATENA. This transcriptional increase was accompanied by increased BG4-binding at this G4-site, as quantified by BG4 CUT&Tag qPCR, suggesting that PDS binding might enhance local protein accessibility, acting as a molecular glue and, thereby, stimulating transcription. In contrast, PhenDC3 treatment resulted in transcriptional suppression—consistent with the more commonly reported effects of G4 ligands-demonstrating that structurally distinct ligands can drive divergent transcriptional responses at the same G4 site. Notably, these ligand-dependent responses were also observed when using the free—not bound to dCas9—PyPDS or PhenDC3, indicating that the lack of inter-G4 selectivity does not necessarily prevent these molecules from providing meaningful information on the specific G4 site. These findings suggest that G4 ligands—while useful tools for perturbing G4 homeostasis and investigating the consequent biological responses

−cannot be used to directly infer the native biological roles of G4s but should instead be used to gain insights into the responses triggered by their binding to these structures. Moreover, our data indicate that it is not safe to assume that different G4 ligands will lead to similar biological responses, which is a common assumption often reported in the literature. While we showcase the application of ATENA for studying the transcriptional responses associated with G4- and iM-targeting, it is important to acknowledge that the cellular responses linked with G4 targeting by ligands are not limited to transcriptional modulation. For example, treatment with PDS and PhenDC3 has been linked to several cellular pathways that contribute to their overall therapeutic potential. Indeed, the transcriptional modulation linked with PDS treatment was initially discovered through its ability to trigger a DNA damage response at G4 sites[88]. It has been successively revealed that other G4 ligands can trigger a DNA damage response through G4 targeting, typically in a replication and transcription-dependent fashion, representing a significant aspect of G4 ligand activity that operates independently of direct transcriptional effects[55,56,89,90]. Additionally, the cellular activity of G4 ligands has been associated with telomere instability, where ligand binding to telomeric G4 structures can disrupt telomerase function and compromise telomere maintenance[46]. The complexity of these interactions underscores the multifaceted nature of G4-ligand biology and highlights that, while important, these transcriptional effects represent only one aspect of their cellular activity.

Therefore, while ATENA enables precise dissection of transcriptional responses attributable to individual G4-targeting, we emphasize that the broader therapeutic potential of G4-ligands likely stems from the integration of multiple molecular pathways. Future applications of ATENA could be expanded to investigate these alternative mechanisms by coupling selective G4 targeting with assays for DNA damage, telomere stability, or other cellular processes, thereby providing a more comprehensive understanding of G4 ligand biology.

Finally, we utilized ATENA to target uncharacterized G4s previously identified in MCF7 cells with CUT&Tag but not investigated for their biological functions. Given that G4s are typically found at active promoters, we reasoned that a gene's transcriptional status can influence its responsiveness to G4-ligand targeting. We took advantage of ATENA to direct PyPDS to a G4-promoter of either a highly expressed gene (*HMGN1*) or a lowly expressed one (*IL17RA*). Single targeting of the G4 located in the *HMGN1* promoter−unique to MCF7 cells and not detected in other cancer cell lines[76]−led to the greatest transcriptional suppression measured in MCF7 (a 99% reduction), confirming a strong link between transcriptional activity and ligand sensitivity that indicates a substantial contribution of the HMGN1-G4 in sustaining elevated expression levels of *HMGN1* in breast cancer cells.

We recently observed that PDS treatment in chemo-resistant ovarian cancer cells resensitizes them to chemotherapy, due to the enrichment of G4s in highly transcriptionally active genes that are essential to establish a chemo-resistant state, such as *WNT*[91]. Given the relevance of epigenetic changes in establishing dormancy in breast cancer[81] and our observation linking the expression levels of *HMGN1* with the dormancy/awakening transition (Supplementary Fig. 10), targeting HMGN1-G4 with ligands might offer scope for the development of a therapeutic strategy to prevent the awakening of dormant cells and cancer relapse.

Altogether, these results support a model in which the biological significance of G4 structures within the genome is contingent on the epigenetic landscape and the gene transcriptional status in a given cell line.

In conclusion, ATENA provides a robust method for targeting single G4s within the genome of living cells using distinct G4-ligands. This technology allowed us to dissect the diverse biological responses elicited by different ligands at the same G4 site, as well as the effects of targeting various G4s with the same ligand with high precision. We found that the chemical nature of the ligand used can perturb the local protein-binding homeostasis of promoter G4s differently, leading to either increased protein accessibility and transcriptional activation or decreased accessibility and transcriptional suppression. These effects can be monitored directly using BG4 CUT&Tag qPCR, providing an assessment of protein accessibility at the targeted G4 as a response to ligand binding. We demonstrated that some ligands could act as molecular glues of G4-protein interactions or as displacers, depending on the specific G4 targeted. This can be valuable for developing therapeutic agents based on G4 targeting, which can be tailored chemically to either diminish or amplify transcription based on their binding modalities. Moreover, ATENA enabled us to disentangle local effects triggered by individual G4 targeting from broader responses driven by global G4 stabilization obtained when using canonical G4 ligands. Interestingly, our data suggest that G4 ligands−despite their lack of inter-G4 selectivity−often recapitulate the transcriptional changes observed with the selective targeting of G4s and can be used to infer the therapeutic potential of individual structures. Indeed, the response to G4-ligand treatment is mainly shaped by the intrinsic level of transcription of the targeted promoter, rendering ligand responses highly epigenetic and cell-dependent.

Despite these advantages, we acknowledge that ATENA also presents inherent technical limitations that must be considered for its broader application. For example, ATENA relies on available PAMs near a structure of interest that may limit access to certain G4s or iMs. Additionally, potential steric hindrance effects from the dCas9-Halo/ligand complex and the current requirement for individual sgRNA optimization for each target present practical challenges for high-throughput applications. However, several promising avenues exist for addressing these limitations, particularly the issue with the PAM constraint. Cas variants such as chimeric Cas9 enzymes offer more relaxed PAM requirements and could significantly expand ATENA's applicability and its wider scope[92]. Additionally, the development of near PAM-less Cas9[93] and CRISPR variants represents exciting opportunities for reducing or eliminating PAM dependencies, substantially broadening ATENA's applicability in the coming years while retaining its precision and modularity.

We envisage that further development of ATENA can be leveraged to generate genome-wide screening of G4-functional responses to various ligands in different cell lines. Additionally, this technology can be used to generate Halo-functional ligands and screen their ability to perturb protein accessibility at a given G4 site. We also anticipate that−given the minimal perturbation of ligands required for Halo-functionalization (chloroalkane)−ATENA is potentially a promising platform for gene-selective localization of other DNA-interacting therapeutics, including topoisomerase inhibitors or cross-linking agents.

## Methods

Comprehensive synthetic protocols and purification methods for Cl-PyPDS$_n$, Cl-PhenDC3$_n$, and Cl-pep-RVS$_n$ ligands are provided in the Supplementary Information 1.

### FRET-melting assay

Before each experiment, a solution containing 20 μM DNA ordered from Integrated DNA Technologies (IDT) (Supplementary Table 1), labeled at 5′ with FAM and at 3′ with TAMRA, was freshly prepared in 10 mM lithium cacodylate buffer, pH 7.4 supplemented with 20 mM KCl. For the non-G4-forming sequence dsDNA26 (26-mer), the solution was supplemented with 150 mM KCl. The DNA solution was placed on an Eppendorf Thermomixer, annealed by heating at 95 °C for 10 min, and then cooled at a rate of 0.5 °C/min. The DNA was aliquoted into a 96-well RT-PCR plate (Thermo Fisher Scientific) and supplemented with increasing equivalent amounts of each ligand, resulting in a final oligonucleotide concentration of 0.4 μM. Fluorescence readings were

performed on an Agilent Stratagene MX3000P, in the range of 25-95 °C, in technical duplicate. The data points obtained were analyzed with GraphPad Prism 10, and the melting temperature was extrapolated from a Boltzmann sigmoidal function. Statistical significance was calculated using a two-tailed t-test in GraphPad Prism. $p$-value: ns > 0.05, *≤0.05, **≤0.01, ***≤0.001, ****≤0.0001.

## CD-melting assay

Circular Dichroism (CD) melting experiments were conducted on an AVIV Biomedical Inc. (Lakewood, NJ, USA) Model 410 Circular Dichroism Spectrometer. Before each experiment, c-MYC-Pu22 (Supplementary Table 1) was annealed at a concentration of 2 μM in lithium phosphate buffer (pH 7.4, 20 mM phosphate) supplemented with 1 mM KCl. When required, the ligand was added to the annealed solution and equilibrated for 1 h before initiating the melting process.

Each experiment was performed in triplicate using a 10 mm optical path length quartz cuvette, with a 2 mm bandwidth. The temperature was ramped from 30 °C to 95 °C, with scans recorded every 5 °C after a 2 min equilibration time. CD signals were measured at 264 nm (corresponding to the parallel G-quadruplex maximum).

The CD values (mdeg) at 264 nm were plotted as a function of temperature, normalized, and fit to a Boltzmann sigmoidal equation using GraphPad Prism 10.0. This analysis yielded the melting temperature (V50) of the G4 structure in the presence of increasing ligand equivalents. Statistical significance was calculated using a two-tailed t-test in GraphPad Prism. p-value: ns > 0.05, *≤0.05, **≤0.01, ***≤0.001, ****≤0.0001.

The CD melting experiments for the Cl-pep-RVS$_4$ peptide were performed using a Jasco J-1500 spectropolarimeter with 10 μM DNA samples in 10 mM sodium cacodylate buffer at pH 6.6, in a 1 mm path length quartz cuvette. Two repeats of CD melting full spectrum ranges were taken (from 230 to 320 nm) for the long C-rich *c-Myc* sequence (Supplementary Table 1), measuring the unfolding of the DNA structures from 5 to 95 °C in the presence of 10 molar equivalents of DMSO or Pep-RVS. The samples were kept at 5 °C for 5 min before starting the melt with 1 °C/min increase in temperature, 0.5 °C data interval, and 60 s holding time at each target temperature. Four scans were accumulated with a data pitch of 0.5 nm, a scanning speed of 200 nm/min, 1 1-s response time, a 2 nm bandwidth, and a 200 mdeg sensitivity. The data were zero-corrected to 320 nm, and baseline drift was corrected. The melting temperature (TM) was determined using the Biphasic curve fitting on the normalized data using GraphPad Prism version 10.1.2. Data were processed as mean ± SEM ($n = 2$) and one-way ANOVA followed by Bonferroni post-hoc test to determine significant changes between peptides and controls.

## UV-binding assay

Oligonucleotides c-MYC-C52 and c-MYC-G52, purified by reverse-phase HPLC, were purchased from Eurogentec (Supplementary Table 1). The oligonucleotide stock solutions were dissolved in Milli-Q water and diluted to a final concentration of 250 μM in stock buffer (10 mM sodium cacodylate, pH 6.6) and annealed by heating at 95 °C for 5 min and cooling to room temperature overnight. The peptide was dissolved to 10 mM in DMSO, and 1 mM in DMSO was used in the experiment. UV-based binding curves were determined using the wavelength at 350 nm for 10 μM pep-RVS or 10 μM Cl-pep-RVS$_4$ in stock buffer. DNA (250 μM in stock buffer) was added in increments up to a 50 μM final concentration.

## dCas9-Halo purification

The expression vector for dCas9-Halo (Addgene #72269) was introduced into BL21 cells (D3, NEB-C2527H) by heat shock. Bacteria were then plated on LB agar plates and selected with Ampicillin (100 μg/mL). One bacterial colony was picked and grown overnight in a 50 mL flask at 37 °C. Overnight precultures were diluted 1:100 into the main

culture (LB medium, Ampicillin 100 μg/mL) and incubated at 37 °C, 200 rpm until an OD$_{600}$ of 0.6. The expression of the recombinant dCas9-Halo was induced by using 1 mM isopropyl β-D-1-thiogalactopyranoside (IPTG). Upon induction, the temperature was reduced to 16 °C for dCas9-Halo expression. BL21-dCas9-Halo cells were harvested after overnight incubation at 16 °C by centrifugation (3500 g, 20 min, 4 °C). Cells were lysed in lysis buffer (50 mM sodium phosphate, 300 mM NaCl, pH 7.2, 20% glycerol, 0.1% Tween-20, 1 mM TCEP, 1 mg/mL lysozyme, 1 mM PMSF, 1 mM MgCl$_2$, and 5 U/g of culture Benzonase). Cell lysates were then homogenized, and cell debris was removed by centrifugation (16,000 g, 15 min, 4 °C). The soluble fraction of the lysate was filtered and then loaded onto a cobalt column, and the bound protein was eluted with 150 mM imidazole. After imidazole removal and concentration by using a 50,000-MWCO centrifugal filter (Millipore Amicon), fractions were loaded onto a 1-mL HiTrap SP XL (GE Healthcare), ÄKTA pure 25 for cationic exchange chromatography, and elution was performed with a gradient from 0% to 70% of buffer (50 mM HEPES pH 7.5, 1 M KCl, 1 mM TCEP). The purest fractions were pooled and then used for gel filtration chromatography. Gel filtration was performed using a Superdex 200 10/300 GL column, and isocratic elution was performed in a buffer containing 50 mM HEPES (pH 7.5), 150 mM KCl, and 1 mM TCEP. Protein aliquots in storage buffer (50 mM HEPES, pH 7.5, 150 mM KCl, 20% glycerol) were snap-frozen and stored at -80 °C.

## BG4 purification

BL21 *E. coli* competent cells (D3, NEB-C2527H) were transformed with a pSANG10-3F-BG4 plasmid (Addgene #55756). Single colonies were inoculated in 2xYT medium supplemented with 2% glucose and 50 μg/mL kanamycin and incubated at 30 °C at 250 rpm overnight. The starting culture was diluted 1:500 into ZYM-5052 autoinduction medium prepared following the previously described method[94] supplemented with 50 μg/ml kanamycin. The culture was then incubated at 37 °C for 6 h at 250 rpm, followed by an overnight incubation at 25 °C at 280 rpm. The culture was centrifuged at 4000 g, 4 °C for 30 min, and the collected pellet was lysed in 80 mL ice-cold TES buffer (50 mM Tris-HCl, pH 8.0, 1 mM EDTA, pH 8.0, 20% sucrose) and incubated on ice for 10 min. This was followed by 15 min of incubation in ice-cold TES (diluted 1:5), supplemented with 15 U/mL benzonase, 2 mM MgSO$_4$, and a protease inhibitor cocktail. The lysate was centrifuged at 16,000 g for 20 min. The supernatant was collected and incubated with nickel resin (Sigma, P6611), previously equilibrated in PBS, at 4 °C for 1 h with rotation. The complex was washed three times with wash buffer (PBS, 100 mM NaCl, 10 mM imidazole). Then, BG4 was eluted in 4 mL of elution buffer (PBS, pH 8.0, 250 mM imidazole) and dialyzed against PBS at 4 °C using GeBaflex tubes (Generon). Purified BG4 was snap-frozen in liquid nitrogen and stored at −80 °C.

## dCas9-binding competition assay

dCas9-Halo purified protein (4 μM) was incubated with several dilutions of Cl-PyPDS$_n$ or Cl-PhenDC3$_n$ probes or with commercially available Halo-TAMRA for the positive control (Promega, G8252) for 30 min at room temperature in PBS. Pretreated dCas9-Halo proteins were then incubated with 5 μM Halo-TAMRA, or Cl-PyPDS$_n$ for the positive control, and samples were loaded on an SDS-PAGE gel (8% polyacrylamide) and run at 180 V for 45 min. Gels were then imaged using a Typhoon FLA 9500 (GE) with the TAMRA filter set (excitation at 542 nm) and then stained with Coomassie. Images of Coomassie-stained gels were then acquired with ImageQuant LAS 4000 (Cytiva).

Each band intensity was then quantified using Image Studio software and normalized for the corresponding Coomassie signal, expressed as a percentage (with 100% labeling corresponding to the positive control). IV-CP$_{50}$ values were determined using nonlinear regression (dose-response inhibition curves with constrained fitting) in GraphPad Prism with $n$ = (number of independent experiments) = 2.

### In vitro transcription for sgRNA synthesis

Oligonucleotides containing sgRNA sequences (Supplementary Table 2) were ordered from Integrated DNA Technologies (IDT), and a PCR was set up to generate the corresponding duplex following the previously described method[95]. Following the manufacturer's instructions, the DNA product was transcribed using the HiScribe T7 High Yield RNA Synthesis Kit (NEB, E2040S). The resulting sgRNA was purified using the RNeasy Mini kit (74104) and stored at −80 °C.

### dCas9-Halo FRET assay

Before the experiment, Cy5-3' end-labeled and Cy3-5' end-labeled oligos (Supplementary Table 1) were annealed in 10 mM Tris-HCl, pH 7.5, 100 mM KCl at a final concentration of 1 μM (95 °C for 10 min and left for overnight cooling to room temperature). dCas9-Halo purified protein at a final concentration of 4 μM was incubated with Cl-PyPDS$_n$ probes in a 1:2.5 ratio for 45 min in binding buffer (20 mM HEPES, pH 7.5, 100 mM KCl, 1 mM MgCl$_2$, 1 mM TCEP, 10% glycerol). The pre-treated dCas9-Halo was then incubated with 120 pmol of purified sgRNA for 20 min at room temperature, followed by incubation with 200 nM of pre-annealed oligos at 37 °C for 1 h. The complexes were then loaded into an 8% polyacrylamide native gel and run for 120 min at 4 °C in 1x TBE supplemented with 2 mM MgCl$_2$. Gels were then imaged using a Typhoon FLA 9500 (GE) with the Cy3 and Cy5 filter sets. Analysis of the fluorescent band intensity (I) and relative FRET efficiency (E) was performed using a Python code. Particularly, FRET efficiency was calculated as: $E = I_{Cy5} /(I_{Cy3} + I_{Cy5})$. The signals in both channels were normalized for the background and the sgRNA NTC control, and for every sgRNA, the ΔFRET is defined as (E+ligand) - (E −ligand). Statistical significance was calculated using a two-tailed $t$ test in GraphPad Prism; $p$-value: ns > 0.05, *≤0.05, **≤0.01, ***≤0.001, ****≤0.0001 with $n$ = (number of independent experiments) = 2.

### Cloning of lentiviral dCas9-Halo construct

The lentiviral dCas9-Halo plasmid was cloned using Gibson assembly. A lentiviral backbone containing dCas9 (Addgene #61425) was digested using BamHI and BsrGI (New England Biolabs, R3575S, R3136S) and then assembled with the HaloTag sequence amplified (Table.S3) from pET302-6His-dCas9-Halo (Addgene #72269) in a 1:3 molar ratio of backbone: insert using HiFi DNA Assembly Mix (E2621S, New England Bioscience) following manufacturer's protocols. Post-incubation, assembled products were diluted with water, and 5 μL of the product was transformed by heat shock into 10-beta competent cells (New England Biolabs, C3019I). Cells were then plated on agarose plates (supplemented with ampicillin 100 μg/mL) for overnight outgrowth at 37 °C. Single clones were picked and grown in 5 mL of Amp LB medium overnight at 37 °C while shaking. Plasmid DNA was purified from cells using the Promega PureYield plasmid miniprep system (Promega, A1223) and sequenced using the Genewiz service.

The positive plasmid was further modified by substituting the blasticidin resistance gene with the mCherry coding sequence using Gibson assembly after digestion with BsrGI and EcoRI enzymes (Supplementary Table 3). Positive clones were then sequenced using the Genewiz service.

### Cloning of sgRNA in pLG1 plasmid

The pLG1 backbone (Addgene #109003) was digested with BstXI and BlpI (FastDigest, ThermoFisher) for 1 h at 37 °C. Oligos containing the sgRNA sequence (Supplementary Table 4) were ordered from IDT and annealed in water at 10 μM in a thermocycler (37 °C, 30 min −95 °C, 5 min; and ramp down 5 degrees/min to 25 °C). The annealed oligos were diluted (1:50) and then assembled with the digested plasmid in a 1:2 molar ratio of backbone: insert using HiFi DNA Assembly Mix (E2621S, New England Bioscience) following the manufacturer's protocols. Post-incubation, the assembled products were diluted 1:2 in water, and 5 μL of the product was transformed by heat shock into

10-beta competent cells (New England Biolabs, C3019I). Cells were then plated on agarose plates (supplemented with 100 μg/mL ampicillin) for overnight outgrowth at 37 °C. Single clones were picked and grown overnight in 5 mL of Amp LB medium at 37 °C in a rotating shaker. Plasmid DNA was purified from cells using the Promega PureYield plasmid miniprep system (Promega, A1223) and sequenced using the Genewiz service.

### Mammalian cell culture

HEK293FT cells were cultured in DMEM GlutaMAX™ (Thermo Fisher Scientific, 10566016) supplemented with 1x penicillin-streptomycin (Thermo Fisher Scientific, 15140122) and 10% (vol/vol) fetal bovine serum (FBS) and maintained at 37 °C and 5% CO$_2$.

MCF7 cells were cultured in DMEM GlutaMAX™ (Thermo Fisher Scientific, 10566016) supplemented with 1x β-Estradiol (Sigma, 50-28-2), 1x penicillin-streptomycin (Thermo Fisher Scientific, 15140122), and 10% (vol/vol) fetal bovine serum (FBS) and maintained at 37 °C and 5% CO$_2$.

### Lentivirus production and transduction

For lentiviral production, HEK293T cells were seeded at $3.8 \times 10^6$ in a 10 cm tissue culture plate the day before transfection. 8.4 μg of the envelope plasmids pCMV-VSV-G (Addgene #8454) and 6.4 μg of packaging plasmid R8.74 (Addgene #22036) were co-transfected along with 2.1 μg of the target plasmid (dCas9-Halo-T2A-mCherry) using polyethyleneimine (PEI-Linear, MW 25,000) in a μg DNA: μg PEI ratio of 1:3. Viral supernatant was harvested after 48 h and 72 h, and before usage, it was filtered using a 0.45 μm filter unit. MCF7 cells were plated on 6-well plates the day before infection. They were infected with lentiviruses in DMEM, 10% FBS, β-Estradiol, in the presence of polybrene with a final concentration of 10 μg/mL. Half of the medium was changed the day after, and cells were grown for one week and examined by flow cytometry (Attune NxT, ThermoFisher) to confirm successful transduction (YL2-Channel). After genotyping (Supplementary Table 3), cells were single-cell sorted using FACS BD FACS Diva 9.0.1 (Supplementary Fig. 11).

### Western blot

MCF7 cells stably expressing dCas9-Halo were harvested and processed using 1x RIPA buffer (Merck,20-188) supplemented with 1x Protein inhibitor cocktail. The total protein concentration of the cleared lysate was then measured by Bradford assay (Pierce™ Bradford Plus Protein Assay Kits, Thermo Fisher Scientific, 23236) at 595 nm using a ClarioStar plate reader. A total of 20/25 μg protein was then loaded into an 8% polyacrylamide SDS-PAGE gel and transferred to a pre-assembled PVDF membrane (Trans-Blot Turbo Mini 0.2 μm PVDF Transfer Packs, Bio-Rad) using a semi-dry method (Bio-Rad). The membrane was blocked (5% milk in TBS-T) and incubated with primary antibodies (1:1000 Cas9 antibody- 14697 T, 1:1000 GAPDH antibody- 2118 T, Cell Signaling Technology) overnight at 4 °C. The membrane was then washed in TBS-T and incubated with secondary antibodies (1:10,000) goat anti-rabbit/mouse HRP-Advansta, R-05071-500, R-05072-500) for 1 h at room temperature. After three washing steps of the membrane in TBS-T, 0.5 mL of HRP substrate (Merck, WBLUC0100) was added to the top of the membrane, and the excess was removed. The signal was developed using Image Quant LAS 4000 (Cytiva) with the following settings: chemiluminescence for the WB signal and Cy5 for visualizing the protein marker.

### CAPA assay

MCF7 cells stably expressing dCas9-Halo were plated the day before at $30 \times 10^3$ cells/ well in 96-well plates pre-coated with Poly-D-Lysine (Thermo Fisher Scientific, A3890401). Cells were incubated for 2 h in the presence of serial dilutions of Cl-PyPDS$_n$, Cl-PhenDC3$_n$, or Cl-pep-RVSn probes ranging from 0.2 μM to 10 μM (5% CO$_2$, 37 °C). Cells were

then washed two times with medium (every washing step included 10 min of incubation, 5% $CO_2$, 37 °C) and incubated with HaloTag® Oregon Green® Ligand (Promega, G2801) for 45 min, followed by two washing steps (10 min, 5% $CO_2$, 37 °C). Lastly, cells were washed with DPBS, treated with 0.25% trypsin, and resuspended in FACS buffer (5% BSA in DPBS) before undergoing flow cytometry analysis (Attune Nxt). Data analysis was performed in FlowJo 10.9.0. Mean fluorescence values from two biological replicates (each with three technical replicates) were normalized to the positive-control signal and expressed as percent labeling. Following this, $CP_{50}$ values were obtained by non-linear regression (dose–response inhibition curves with constrained fitting) in GraphPad Prism ($n = 2$).

## Transfection and incubation with G4 ligands
MCF7 cells stably expressing dCas9-Halo were plated the day before at $25 \times 10^3$ cells/well in 96-well plates pre-coated with Poly-D-Lysine (Thermo Fisher Scientific, A3890401). According to the manufacturer's protocol, 50 ng of guide-expressing plasmid was transfected using Lipofectamine 3000 (Thermo Fisher Scientific, L3000001). The day after, cells were incubated with either DMSO (0.5%) or Cl-PyPDS$_n$, Cl-PhenDC3$_n$, or Cl-pep-RVS$_n$ probes and incubated for 48 h.

## RNA isolation
RNA was harvested 72 h post-transfection. Cells were washed with 100 µL of 1X DPBS (Gibco, 14190144) and incubated with RLT buffer from RNeasy mini kits (Qiagen, 74104) according to the manufacturer's instructions. Following the manufacturer's instructions, the eluted RNA was reverse-transcribed into cDNA using the RevertAid cDNA Synthesis Kit (K1621, Thermo Fisher Scientific).

## Quantitative real-time PCR (RT-qPCR)
RT-qPCR reactions were prepared as follows: cDNA was mixed with 500 nM of forward and reverse primer (Supplementary Table 5) and 5 µL of Fast SYBR Green Master Mix (Applied Biosystems: 4385612) in a final volume of 10 µL. RT-qPCR was performed on a 96-well plate format using an Agilent Stratagene Mx3000P machine with the following program: 95 °C for 20 s, 40× (95 °C for 3 s, 60 °C for 30 s), 95 °C for 15 s, 60 °C for 1 min, and 95 °C for 15 s (melting curve). The $C_t$ values obtained were used to calculate the relative fold-change in gene expression using the $\Delta\Delta C_t$ method and normalized to the housekeeping gene *GAPDH* in both treated and untreated samples. Statistical significance was calculated using a Welch-corrected two-tailed t-test in GraphPad Prism; *p*-value: ns > 0.05, *≤0.05, **≤0.01, ***≤0.001, ****≤0.0001.

## BG4 CUT&Tag
Cells were plated in a 6-well plate format one day before transfection, following the previously described transfection protocol. 24 h after transfection, cells were incubated with Cl-PyPDS$_n$ probes for 6 h and then harvested, counted, and inspected for viability. To prepare nuclei, 200,000 cells per reaction (plus 10% excess) were centrifuged for 3 min at 600 $g$ at room temperature and then resuspended in nuclei extraction buffer (20 mM HEPES–KOH, pH 7.5, 10 mM KCl, 0.1% Triton X-100, 20 mM Glycerol, 0.5 mM Spermidine, 1X Roche cOmplete Mini EDTA-free Protease Inhibitor (Roche, 11836170001) and incubated for 10 min on ice. Nuclei were spun at 600 x g for 3 min at 4 °C and then gently resuspended in 200 µL/per reaction cold Nuclei Extraction Buffer. A 2 µL aliquot was taken and mixed with 2 µL trypan blue to examine nuclei integrity under the microscope. In the meantime, ConA beads (BioMag®Plus Concanavalin A, BP531) were prewashed and equilibrated in binding buffer (20 mM HEPES–KOH, pH 7.5, 10 mM KCl, 1 mM $CaCl_2$, 1 mM $MnCl_2$). Nuclei for each condition that needed to be tested were mixed with 10 µL of beads and incubated for 10 min at RT while shaking. After checking the binding of the beads to the ConA beads under the microscope (2 µL sample + 2 µL trypan blue), the

samples were washed two times with 300 µL of 1% BSA antibody buffer (20 mM HEPES–KOH, pH 7.5, 150 mM KCl, 0.5 mM Spermidine, 1X Roche cOmplete Mini EDTA-free Protease Inhibitor, 5% Digitonin, 2 mM EDTA, and 1% BSA). The tubes were placed on the magnetic rack, and the liquid was withdrawn. Samples were then resuspended in 50 µL/ reaction of 1%BSA antibody buffer and incubated for 1 h at RT while shaking. The samples (3 technical replicates for each condition) were incubated with 216 nM of an in-house prepared BG4 antibody and 0.5 µL of H3K27me3 (Cell Signaling Technology, 9733S) antibody for the positive control (1 replicate for each condition) and left overnight at 4 °C while shaking.

The day after, samples were checked to ensure that they were bound to the beads and then washed two times with Dig-wash buffer (20 mM HEPES–KOH, pH 7.5, 150 mM KCl, 0.5 mM Spermidine, 1X Roche cOmplete Mini EDTA-free Protease Inhibitor, 5% Digitonin). Samples containing BG4 and samples for the negative control were resuspended in 50 µL of Dig-wash buffer containing 2 µL of anti-flag antibody (DYKDDDDK Tag Antibody, 2368S) and incubated for 1 h at RT while shaking. Then, the BG4 and negative samples were washed three times in 100 µL Dig wash buffer, resuspended in 50 µL Dig wash buffer supplemented with 0.5 µL of tertiary antibody (anti-rabbit, Abcam, ABIN101961) and incubated for 1 h at RT while shaking. Samples were then washed three times with 100 µL Dig-wash buffer, resuspend in 50 µL Dig-300 buffer (20 mM HEPES–KOH, pH 7.5, 300 mM KCl, 0.5 mM Spermidine, 0.01% Digitonin, 1X Roche cOmplete Mini EDTA-free Protease Inhibitor) containing pA-Tn5 (CUTANA™ pAG-Tn5 for CUT&Tag, 15-1017) adapter complex (1:20) and incubated for 1 h at RT while shaking. Samples were then washed three times in Dig-300 buffer, resuspended in 100 µL of Tagmentation buffer (Dig-300 buffer supplemented with 10 mM $MgCl_2$) and incubated for 1 h at 37 °C. Samples were then washed twice with TAPS buffer (10 mM TAPS buffer- J63268.AE, Thermo Scientific Chemicals, 0.2 mM EDTA), mixed by vortexing in 100 µL of Protenaise K buffer (0.5 mg/mL Proteinase K, Thermo Fisher Scientific EO0492/EO0491, 0.5% SDS in 10 mM Tris-HCl buffer pH 8.0) and incubated 1 h at 55 °C. Samples were then purified with Zymo DNA Clean & Concentrator-5 (D4013) and samples were then amplified using NEBNext HiFi 2x PCR Master mix (NEB, M0541) with the following set up (Initial 72 °C for 5 min, 20X cycle (98 °C for 30 sec, 98 °C for 10 sec, 63 °C for 10 sec) and 72 °C for 1 min). Samples were then purified using Ampure XP beads (A63880) according to the manufacturer's instructions. Quality control of the sample was performed using TapeStation Screen-Tape HSD1000 (5067-5584) and TapeStation ScreenTape HSD1000 reagents (5067-5585) according to the manufacturer's instructions on a TapeStation (TapeStation Agilent 4150).

## BG4 CUT&Tag-qPCR
CUT&Tag libraries were diluted 1:10, and 2 µL of the diluted CUT&Tag library was mixed with 1 µM of primer mix (Supplementary Table 6) and 5 µL of SYBR Green PCR Master Mix (Applied Biosystems: 4385612) according to the following protocol 20 s at 95 °C, 39 cycles of 10 s at 95 °C, 30 s at 60 °C, and 10 s at 72 °C, and finally heating to 90 °C. The $C_t$ values obtained were used to calculate the relative fold-change in gene expression using the $\Delta\Delta C_t$ method and assess the relative fold change at G4 target sites when compared against G4-positive regions (MAZ, RBBP4, RPA3). Statistical significance was calculated using a Welch-corrected two-tailed *t* test in GraphPad Prism; *p*-value: ns > 0.05, *≤0.05, **≤0.01, ***≤0.001, ****≤0.0001.

## Histone modifications CUT&Tag
The CUT&Tag assay was performed using the EpiCypher CUTANA Direct-to-PCR CUT&Tag Protocol v1.7, which included all the mentioned materials and buffer recipes with minor modifications. Briefly, cryopreserved nuclei were thawed quickly and immobilized to Concanavalin A (ConA) Conjugated Paramagnetic Beads (Epycipher, cat.

21-1401). Bead-bound nuclei were spiked with 2 µL of spike-in SNAP-CUTANA™ K-MetStat Panel (1:50, Epicypher, cat. 19-1002) and immediately resuspended and incubated in Primary Antibody diluted in Antibody buffer at the manufacturer's CUT&Tag recommended dilutions, at 4 °C by nutation overnight. The next day, the primary antibody was removed, and nuclei were incubated with either anti-Rabbit or anti-Mouse Secondary Antibodies at Room Temperature (RT) in Digitonin 150 buffer for 1 h with nutation. Nuclei were washed twice with wash 150 buffer and incubated with CUTANA pAG-Tn5 enzyme (Epicypher, cat. 15-1017) diluted in Digitonin 300 buffer for 1 h at RT with nutation and thoroughly rewashed twice. Nuclei were resuspended in Tagmentation Buffer containing MgCl2 and incubated 1 h at 37 °C. The reactions were then washed with Post-tagmentation buffer and incubated in SDS Release Buffer at 58 °C for 1 h to quench the tagmentation reaction. Then, SDS Quench Buffer was added to neutralize SDS, as well as NEBNext High-Fidelity 2 × PCR Master Mix (NEB, cat. M0541L), and 2.5 µM of universal i5 primer and unique i7 primer sequences previously published in Buenrostro et al. 2015 (IDT technologies). Libraries were amplified at 14 cycles for abundant histone modification samples (H3K27me3), and 16 cycles for less abundant marks (H3K4me3) and negative control IgG following the cycling parameters outlined in the EpiCypher® CUTANA™ Direct-to-PCR CUT&Tag Protocol v1.7. Library clean-up was performed using 1.3x SPRI beads (Beckman Coulter, cat. B23319) to recover ~75 bp DNA fragments. The beads-DNA were incubated at RT for 5 min, followed by two washes with 85% ethanol. Libraries were eluted in 15 µL of 0.1x TE buffer, quantified using Qubit™ fluorometer per manufacturer's instructions, and quality controlled for fragment length enrichment using the Agilent TapeStation Bioanalyzer® with High Sensitivity D1000 reagents. The following antibodies were used for CUT&Tag: H3K4me3 (Epycypher, cat 13-0041, 1:50), H3K27me3 (Cell Signalling, cat 9733T, 1:50), IgG (Epycypher, cat 13-0042, 1:1000), Anti-Rabbit Secondary Antibody (Epycypher, cat 13-0047, 1:100) and Anti-Mouse Secondary Antibody (Epycypher, cat 13-0048, 1:100). Libraries were sequenced at 10 million reads per sample on Paired-End platform of 150 bp per fragment (PE150) by our sequencing provider, Novogene, on a NovaseqX platform from Illumina.

CUT&Tag data were processed on the Imperial College London HPC using an adapted version of the pipeline developed by Dr Ye Xheng from the Steven Henikoff lab, available on https://github.com/clabanillas/cutnTag_processing. This pipeline includes quality control, adapter trimming, alignment, filtering, and peak calling. Peaks for H3K27me3 were called using SEACR with no built-in normalization, against control IgG in stringent mode. Peaks for H3K4me3 were called with IgG control with MACS2 callpeak command (--keep-dup all --nomodel --shift -75 --extsize 150 -q 0.01).

## ATAC-seq

scATAC-seq was performed on a Chromium platform (10x Genomics) using Chromium Single Cell ATAC Reagent Kit' V1 chemistry (manual version CG000168 Rev C) and Nuclei Isolation for Single Cell ATAC Sequencing (manual version CG000169 Rev B) protocols. Nuclei suspensions were prepared to get 10,000 nuclei as target nuclei for recovery. Final libraries were loaded onto a Novaseq 6000 platform (Illumina) to obtain 50,000 reads per nucleus with a read length of 2 × 50 bp.

A pseudobulk of single-cell sequencing readout was prepared by concatenating all reads containing the transposase adaptor sequence barcode per sample. Data processing was done according to the following repository: https://github.com/harvardinformatics/ATAC-seq. Briefly, raw paired-end FASTQ files were subjected to quality control using FastQC, and adapter trimming was performed with FASTP (--detect_adapter_for_pe -l 20). Reads were then aligned to the human genome (T2T-CHM13v2.0) using Bowtie2 with parameters optimized for ATAC-seq (--very-sensitive -X 700). Mitochondrial reads were

removed with SAMtools. Duplicates were removed using PICARD (picard-2.27.4-0/picard.jar). Then multimapped reads were removed with SAMtools (-h -q 30), as well as unmapped, mate unmapped, not primary alignment, reads failing platform, duplicates (-h -b -F 1804), and properly paired reads -f 2 were retained. Reads aligned to black-listed regions were removed. Coordinates were shifted with deeptools alignmentSieve command (--numberOfProcessors max −ATACshift). Normalized genome coverage tracks were generated with the BEDtools genomecov function (--numberOfProcessors max --binSize 10 --normalizeUsing RPGC --effectiveGenomeSize 2786136059) using the effective genome size for T2T. Peaks were called using the MACS2 callpeak command (-q 0.01 --keep-dup all).

## mRNA-Seq sample preparation and bioinformatic analysis

For transcriptome-wide gene expression changes in transfected samples, cells were plated the day before in 6-well plates, and 24 h after transfection, they were incubated with either 0.5% DMSO or 2.5 µM Cl-PyPDS_2 for 6 h. For DC-34/PyPDS-treated samples, cells were plated in a 6-well plate the day before treatment with 5 µM and 2.5 µM of the compound for 6 h. Cells were harvested and washed with 1x DPBS (300 g, 5 min). RNA was then extracted using RNeasy mini kits (Qiagen, 74104) according to the manufacturer's instructions. The quality of the extracted RNA was assessed by TapeStation (High Sensitivity RNA Screen Tape, Agilent) and sent to Novogene service for sequencing. RNA libraries were prepared using Novogene kit for library preparation and then sequenced with Illumina NovaSeq X-plus. CASAVA (version 1.8) was used to perform base calling and Phred score. The quality of the reads was assessed with FastQC (Fastp v.0.23.1). Sequence alignment to the reference genome (GRCh38-NCBI:GCA_000001405.27) was performed using HISAT2. Sorted bam files were generated with (Hisat2 v.2.0.5), and gene counts using FPKM.

Raw count data were then imported into DESeq2 (v1.40.2), and differential gene expression analysis was conducted with three comparisons: (1) the generic drug effect in control (NTC) cells treated with the ligand versus mock (DMSO), (2) the effects of the sgRNA (sgRNA_MYC-19) under mock conditions and data set (2) filtered for NTC. DEGs were identified based on adjusted p-values (FDR < 0.05) and log2 fold-change thresholds ( |log2FC| ≥ 1).

## Cell proliferation assay

MCF7 cells, wild-type or stably expressing dCas9-Halo, were plated one day before the experiment at 24 ×10^3 cells/well in a 96-well plate. Cells were then treated 24 h later with serial dilutions (ranging from 0.25 µM to 10 µM) of Cl-PyPDS_2, Cl-PhenDC3_2, Cl-PDC_2, PyPDS, PhenDC3, PDC, DC-34, or Cl-pep-RVSn and 0.5% DMSO as a negative control. After 48 h, cells were incubated with 20 µL/well of CellTiter96®AQ$_{ueous}$One Solution Reagent for 2 h (5% CO_2, 37 °C), before acquiring the absorbance value at 490 nm using a ClarioStar plate reader. The data (Supplementary Table 7) were fitted to a sigmoidal dose-normalized response curve with a variable slope. Statistical analysis of curve-fit parameters was performed by independently fitting data from separate biological experiments, followed by comparison of the resulting curve-fit parameters using Extra sum-of-squares F test in GraphPad Prism. p-value: ns > 0.05, *≤0.05, **≤0.01, ***≤0.001, ****≤0.0001.

## Reporting summary

Further information on research design is available in the Nature Portfolio Reporting Summary linked to this article.

## Data availability

The RNA-seq data generated in this study have been deposited in NCBI's Gene Expression Omnibus under GEO Series accession code GSE279769. The RNA-seq data used to generate the data reported in Supplementary Fig. 10b are available under GEO Series accession number GSE234171. The processed data used to generate Supplementary Fig. 10a can be

found in Supplementary Data 1. Source data are provided with this paper.

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

## Acknowledgements

The authors wish to dedicate this work to the loving memory of Ling Sum Liu, who has sadly passed away while working on this project as a PDRA in our team at Imperial College. M.D.A is supported by a Biotechnology and Biological Sciences Research Council (BBSRC) David Phillips Fellowship (BB/R011605/1) and is a Lister Institute Research Prize holder (2022). S.P.N. and T.E.M. acknowledge support from the Engineering and Physical Sciences Research Council (EPSRC) (EP/S023518/1). R.X. acknowledges the Leverhulme Trust Research Grant (RPG-2022-184), S.G. is supported by the BBSCR [BB/W016710/1]. G.F. is supported by the EPRSC [EP/S023518/1] and by a CRUK non-clinical training award [CANTAC721\100021]. C.S.C. acknowledges support from the Medical Research Council (MRC). L.M. acknowledges CRUK: DRCPFA-Nov23/100009, J.S.S. and C.R.F are supported by the Intramural Research Programs of the National Institutes of Health, National Cancer Institute (NCI), Center for Cancer Research, Project BC011585. D.G. and Z.W. were supported by a BBSRC grant (BB/W001616/1). E.C. was supported by FWO (12B1923N). We are thankful to Prof Luca Magnani for providing MCF7 cells and for helpful discussions. The authors acknowledge Prof Ramon Vilar, Dr Michele Stasi, and Dr Koustav Pal for insightful discussions. The authors also acknowledge Dr Denise Liano, Dr Aisling Minard, Dr Souroprobho Chowdhury and Dr Anna Di Porzio for initial experimental efforts on this project.

## Author contributions

M.D.A. conceived and supervised the research project. S.P.N. designed the research with M.D.A. and performed all the experiments unless otherwise specified. E.C, L.S.L., R.N., A.J.L., E.F., M.Z., T.E.M. have contributed to synthesizing the G4 ligands. E.C. and A.J.L. perform FRET melting assays. E.C. performed the synthesis and chemical characterization of the pep-RVSn ligands and CD melting experiments. S.G. performed with S.P.N. BG4 CUT&Tag experiments. C.S.C. and L.M. performed and analyzed ATAC-seq and CUT&Tag of histone marks. Z.W. and D.G. advised on i-motif-peptides and D.G. performed binding studies on the pep-RVSn ligands. G.F. helped with chromatin prep. J.S.S. conceived DC-34 and helped edit the manuscript. C.R.F synthesized DC-34 used in this study. M.D.A. wrote the manuscript supported by S.P.N with input from all authors. All authors contributed to critical discussion and data interpretation.

## Competing interests

The authors declare no competing interests.
