## [Transparent Peer Review file · Nature Communications]

Chemically modified CRISPR-Cas9 enables targeting of individual G-quadruplex and i-motif structures, revealing ligand-dependent transcriptional perturbation

Corresponding Author: Dr Marco Di Antonio

Version 0:

Reviewer comments:

Reviewer #1

(Remarks to the Author)

This revised submission represents a substantial improvement over the manuscript initially submitted to Nature Cell Biology. I would like to commend the authors for their meticulous and thoughtful responses to the extensive feedback provided by the three reviewers, as well as for the considerable effort reflected in this work. In particular, the discussion of the results has been significantly refined, and the context for this approach is now much clearer. The expanded analyses of 1) the influence of P1, P2, and the TATA-box on c-Myc expression, and 2) the differential cellular responses when the same G4 is targeted by two distinct ligands, are especially compelling. As a result, the manuscript now meets the eligibility criteria for publication in Nature Communications. However, a few points still warrant further attention: 1/ The inclusion of iM-directed investigations strengthens the study by demonstrating the versatility of ATENA. That said, the introductory sections have not been fully updated to reflect these new findings. While the focus remains heavily G4-centric in many places, a more balanced discussion of the putative biological roles of iMs—particularly in support of the arguments presented in lines 98–102—would be beneficial. Additionally, addressing the ongoing debate regarding the relevance of iMab and thus, the existence of iMs in cells (e.g., DOI:10.1093/nar/gkae531) would further enrich the introduction. 2/ Line 217: the claim that no methods have been used to investigate the difference in biological activity between PDS and PhenDC3 is not entirely accurate. For example, a series of genetic screens have been conducted to address this question (see Durocher et al., 10.1016/j.cell.2020.05.040 for instance). 3/ Statistical analyses: the newly added statistical analyses address the reviewers' requests adequately. However, the choice to consider $p < 0.05$ as statistically relevant requires further justification, as this threshold is not standard practice—even when using a two-tailed t-test. 4/ Assay with free ligands: I disagree with the authors' statement that there is “absolutely no value of running this assay with free ligands that will engage with the G4 independently from the sgRNA used”. In fact, where requested (most importantly for the gene expression profiling studies), these experiments are required not “to systematically compare ligands”, as stated in the responses to the reviewer's comments (and they are correct in stating this) but to better emphasize the difference in the cellular response obtained with guided vs unguided ligands. To be completed. 5/ G4 content and DEGs: The G4 content and/or the direct relationship with Myc regulation of the differentially expressed genes (DEGs) discussed in Figure 3, Extended Figure 4, and Tables S9–S12 must be explicitly provided. 6/ I may have missed it but the influence of ATENA built with CI-PDC2 on c-Myc should be shown (at least as a supporting material). 7/ G4 sequences: A minor but important point: A dedicated table should be provided listing the sequences of the G4s discussed, especially for non-classical G4s (e.g., HMG1, IL17RA). 8/ iM study (starting line 634): while the paragraph on iMs introduces novel findings, it raises several questions due to insufficient background and context on iM biology. Are the observed effects consistent with the putative roles of iMs? Why was iMab-based CUT&Tag (see 10.1093/nar/gkad626) not employed to further support the presented results, similar to the approach used for G4s? Is there a functional correlation between c-Myc's G4s and iMs? If so, how might this be explored? For example, would it be relevant to perform G4 CUT&Tag for ATENA with RVS experiments, followed by iM CUT&Tag for ATENA with PDS experiments? As previously noted, the authors must fully integrate the study of iMs by providing all necessary background and context, both in the introduction and in this dedicated section. 9/ General comment on the discussion: this quite long discussion recapitulates the main findings of this study, but fails in large parts to be a real discussion: for instance, the limitations of the techniques must be better described, the comparisons (advantages and drawbacks) with existing techniques must also be better discussed, etc. Some strong choices have been made all along this manuscript (for example to take BG4 as a representative G4 binding protein, or to use a single iM ligand) that must be better defended here. Also, a better contextualisation of these results in the global G4 and iM biology must be provided, insisting on unprecedented questions that are now open to

investigations and must be addressed. Also, there is a strong bias in the discussion (lines 863-867): the cellular effects of ligands do not solely originate in their ability to modulate transcription but also, as discussed for instance with the aforementioned Durocher's paper, in triggering DNA damage in a G4-mediated manner. This is not an isolated example as many others could be cited including a Tarsounas' paper on PDS for instance (10.15252/emmm.202114501), or even one of the author's articles (10.1038/ncomms14432). Many other possibilities exist, as the cellular activity of G4 ligands (including PDS and PhenDC3) has also been linked to telomere instability, immunomodulation activity (e.g. 10.1093/nar/gkab500) and autophagy disruption (e.g. 10.1093/nar/gkz095), etc. Of course, G4-associated transcriptional control could be involved but many other molecular pathways, which remain to be fully understood, could be involved as well. The discussion should therefore adopt a more nuanced perspective to reflect this complexity. 10/ A final (personal) remark: in response to comment #38, the authors state that they "find it inappropriate for a reviewer to use a non-peer-reviewed article to challenge their findings." First, they are entirely correct, and the reviewer should indeed apologize for this (notwithstanding the relative inelegance of the phrasing). Second, given the importance of Byrd's findings—now published in *Nucleic Acids Research* and thus available for use—this control was both critical and fully justified. As it turned out, the authors addressed it, so all is well. Third, the request was not intended to "challenge" the findings but rather to ask for an additional control to further strengthen them—for the sake of the story. Once these points addressed, it will be my pleasure to support the acceptance of this manuscript for publication in *Nat. Commun.*

Reviewer #2

(Remarks to the Author)

In this study, Nuccio et al. reported a CRISPR-Cas9-based platform termed ATENA for targeting of individual G4s/iMs in living cells involving dCas9-Halo and chloroalkane-functionalized ligands. The authors demonstrated that dCas9-Halo functionalized with G4 ligands/iM-binding peptides can regulate respective gene expressions. Furthermore, the authors showed that ATENA can infer the biological relevance of cell line-specific G4s, by which the expression level of the gene targeted can directly influence the functional response upon ligand treatment.

Overall, it is interesting to incorporate CRISPR with ligands to achieve individual G4/iM-targeting. This work demonstrates that ATENA can affect specific gene expression by targeting their G4/iM, which can be modulated by different ligands. I commend the author's efforts in revising this work, and the resubmission/transfer to *Nature Communications*. Below please find my feedback to the authors' response, and also some additional comments from this work (I see it as a resubmission to a new journal, as there are substantial changes to the original submission). I recommend the authors to revise the manuscript based on the following major and minor comments to strengthen different aspect of the manuscript. Once the authors can fully address the comments, I will be happy to re-review it to consider my recommendation for this work to be published in prestigious journal like *Nature Communications*.

Below please find my comment on the author's response on my previous referee report.

1. OK. By showcasing iM, the method name makes more sense, and also provide some degree of novelty compared to similar works in the literature.
2. OK.
3. OK. Please still present it as one of the key limitations when using this technology in discussion. The PAM site requirement may restrict many gene to be considered, and other targeting approach may be more widely accessible instead, e.g. small molecule, peptide, ASO, aptamer, nanobody, etc.
4. OK. One potential future direction will be then to use other dCas protein with different more promiscuous PAM requirement, or some other proteins with no PAM requirement. The author may consider adding to the discussion/future directions if applicable.
5. OK.
6. OK.
7. OK.
8. OK.
9. OK. A follow up question is that for this opposing effect of G4 ligands, are they also dose-dependent, or the effect only diverge when the dose become much more excess than the G4 target or G4 binding proteins in cells?
10. OK
11. OK
12. OK
13. OK
14. OK
15. OK
16. OK
17. OK
18. OK
19. OK
- 20-27. OK

Below please see additional comments from this revised manuscript.

Major Comments:

1. While the bindings between free ligand and chloroalkane-modified ligand to target of interest are assessed, it is important to also verify the binding of dCas9-Halo-ligand with the target of interest, as dCas9 is large in size, and may affect the binding due to steric effects.
2. The authors mentioned that the C-terminal domain of the Cas9 protein points towards the 3' end of the PAM sequence, which means the ligands will be pointed away from the G4 structure when using NT-sgRNA (as shown in fig. 1g). Given this,

please clarify why NT-sgRNAFRET-21-CI-PDS2 also causes an increase in FRET signal? Please explain it more explicitly. For future, are there any general design principles that the author can summarize for the reader in the discussion.

3. For Fig. 1h, please include the Δ FRET of NT-sgRNAFRET-41.

4. The authors demonstrated that targeting MYC-iM has an opposite effect compared to targeting MYC-G4. However, the underlying mechanism remains unclear – is this due to the characteristics of RVS-pep, or the different properties of iM from G4? It will be good to perform a melting test on RVS-pep to verify whether it stabilizes or destabilizes the iM.

5. Since MYC-iM is located on the opposite strand of MYC-G4, it will also be good to assess with sgRNAMYC+22, which is on the same strand as the iM (NT strand).

6. Fig. 5C shows a non-significant increase in IL17RA expression levels. It might be good to further support the hypothesis that IL17RA is unaffected by using more sgRNAs or other ligands.

7. To further distinguish this work with previous work (PMID: 38961283), please clarify the major differences between the two studies. This clarification will help highlight the novel contributions and significance of the current research within the context of existing literature. The authors should include it as supplementary note.

Minor Comments:

1. The authors used PyPDS in their analogues but named them CI-PDSn. To avoid confusion, please rename them as CI-PyPDSn.

2. Please clarify what NTC refers to in this context, whether it is sgRNA targeting unrelated sequences or scrambled sgRNA.

3. Make sure that the significance levels are displayed for all samples in the graphs, including NTC.

4. In page 11 line 287, please specify which SI figure it is after T-sgRNAFRET-41.

5. For clearer illustration, label the 5' and 3' end of the sequences in Fig. 1g.

6. In page 27 line 652 and page 30 line 730, please confirm whether it is chloroalkene or chloroalkane side chain.

Reviewer #3

(Remarks to the Author)

I am impressed with the substantial re-writing and addition of new experiments. The new version clearly states the broad usability and they have addressed clearly all my concerns.

Version 1:

Reviewer comments:

Reviewer #1

(Remarks to the Author)

Notwithstanding our initial disagreements with the lead author of this article, the revised version now exemplifies a remarkable level of scientific and methodological rigor, effectively addressing several critical and long-standing questions in the field of G-quadruplex (G4) research. The depth of mechanistic insight, combined with the innovative technical approaches outlined in this work, positions it uniquely to elucidate persistent questions surrounding both the biological functions of G4 structures and the cellular mechanisms of G4-targeting ligands. The introduction is now more balanced between G4s and iMs, the discussion about the actual influence of P1 on MYC expression is clear and the use of DC-34 as control understandable. As it stands, the explanation about DEGs (page 24) still lacks the precision needed to be entirely convincing, but the differential influence of MYC G4 versus iM (page 28) is now better contextualized and explained. In this context, it would have been great to test the influence of iM on HMGN1 expression (page 34). The discussion section now offers a more comprehensive and nuanced overview of the distinctions between the current study and the work previously published by Qin et al., with Supplementary Table S13 serving as a valuable resource in this comparative analysis. The discussion is also now broader, thus offering a better contextualization of the ATENA scope in the study of G4-related cellular effects, sounder (with the discussion about the limitations of ATENA), and provides interesting avenues for future research. Globally speaking, this is clearly a beautiful piece of science that deserves to be published in Nat. Commun. (On a personal note, I was truly saddened by the news in the Acknowledgements and send my condolences to Marco di Antonio's group.)

Reviewer #2

(Remarks to the Author)

The authors have addressed most of my comments, and I am recommending it for acceptance in Nature Communications.

One final suggestion for the authors:

1. The authors should try to make the manuscript more concise/organized to improve the readability, despite the amount of newly added data during rounds of revision. This will make the messages clearer, and more people will be able to understand the main takeaways and attempt to use their technology.

Reviewer 1

1. The inclusion of iM-directed investigations strengthens the study by demonstrating the versatility of ATENA. That said, the introductory sections have not been fully updated to reflect these new findings. While the focus remains heavily G4-centric in many places, a more balanced discussion of the putative biological roles of iMs—particularly in support of the arguments presented in lines 98–102—would be beneficial. Additionally, addressing the ongoing debate regarding the relevance of iMab and thus, the existence of iMs in cells (e.g., DOI:10.1093/nar/gkae531) would further enrich the introduction.

We agree with the reviewer and have now expanded our discussion on the putative biological roles of i-motifs (iMs) and recent findings on iMab in both the introduction and the relevant section of the revised manuscript. Overall, we demonstrate ATENA's versatility through direct ligand-mediated iM-targeting, which we believe adds to the mounting evidence supporting the existence and druggability of these structures.

2. Line 217: the claim that no methods have been used to investigate the difference in biological activity between PDS and PhenDC3 is not entirely accurate. For example, a series of genetic screens have been conducted to address this question (see Durocher et al., 10.1016/j.cell.2020.05.040 for instance).

We agree with the reviewer that our original wording might have been misleading. What we intended to convey was that no methods have been developed to investigate ligand differences at individual G4s rather than globally. Studies such as the one mentioned and others have indeed addressed this question through genome-wide approaches. We have now rectified this in the revised text, mentioning that whilst this can be achieved globally through genome-wide studies, it has not been possible to examine these effects at single, specific G4 targets - something that now can be achieved with ATENA (Line 265-286).

3. Statistical analyses: the newly added statistical analyses address the reviewers' requests adequately. However, the choice to consider $p < 0.05$ as statistically relevant requires further justification, as this threshold is not standard practice—even when using a two-tailed t-test.

$p < 0.05$ is a highly common threshold for significance and is considered standard practice in biology and biophysics.

4. Assay with free ligands: I disagree with the authors' statement that there is "absolutely no value of running this assay with free ligands that will engage with the G4 independently from the sgRNA used". In fact, where requested (most importantly for the gene expression profiling studies), these experiments are required not "to systematically compare ligands", as stated in the responses to the reviewer's comments (and they are correct in stating this) but to better emphasize the difference in the cellular response obtained with guided vs unguided ligands. To be completed.

We appreciate the reviewer's clarification regarding this point, which may have arisen from the length and density of the original comment. To clarify our position, we stated that experiments with the free ligands have no value in the CAPA assay, where ligands must be tethered to the Halo-tag by design. Similarly, we questioned their utility in the biophysical assays aimed at

optimising the sgRNA positioning and ligand linker length, where the free ligands would provide the same response regardless of the sgRNA used.

However, we fully agree that comparing free ligands with ATENA is essential for cellular responses and transcriptional perturbation studies, as this represents the primary purpose of developing this technology. This is why, in the original submission, we systematically compared these approaches through transcriptome-wide RNA-seq analysis, examining differences between free PyPDS, the same ligand tethered to ATENA targeting MYC-G4, and DC-34. This analysis has allowed us to quantify the number and nature of differentially expressed genes between these 3 different conditions. Furthermore, when discussing the differential response at the PVT1-G4, we demonstrate that the free PyPDS and PhenDC3 ligands do elicit the same response consistent with those observed using ATENA (reported in Extended Data Fig. 5). These comparisons are important because they demonstrate that free ligand treatment remains valuable for obtaining initial insights into ligand-specific transcriptional perturbation, as we discuss in the manuscript. However, generating RNA-Seq datasets for all ligands under all conditions would be a highly expensive and time-consuming effort that would go beyond the scope of this manuscript, which is what we mean when referring to systematic ligand comparisons.

5. G4 content and DEGs: The G4 content and/or the direct relationship with Myc regulation of the differentially expressed genes (DEGs) discussed in Figure 3, Extended Figure 4, and Tables S9–S12 must be explicitly provided.

We have implemented this information explicitly in the text as requested.

6. I may have missed it but the influence of ATENA built with C1-PDC2 on c-Myc should be shown (at least as a supporting material).

In our study, C1-PDS₂ is introduced later in the manuscript as an additional investigation to assess whether the *PVT1* upregulation was specific to PyPDS. MYC-G4 is initially used simply as a proof-of-concept target to optimise sgRNA positioning and PAM selection for ATENA, which we achieve by using the engineered version of PyPDS and PhenDC3. Nevertheless, we have now also performed the requested experiment examining the influence of ATENA/C1-PDC₂ on *c-myc* expression using sgRNA_{MYC-19} (Figure R1), which revealed the expected downregulation consistent with our findings. Whilst we are presenting these data as part of this rebuttal response, we do not feel that this would provide any additional insights for the optimisation of G4-targeting through ATENA, where MYC-G4 is presented. As acknowledged by the reviewer in point 4, this is not a systematic comparison of ligands but rather a study of biological responses observed at different G4 genomic locations.

Figure R1 RT-qPCR for P1-driven *c-MYC* expression in MCF7 cells stably expressing dCas9-Halo transfected with the indicated sgRNAs and incubated for 48h in the presence of (2.5 μ M) Cl-PDC₂. The expression values are represented as fold change ($2^{-\Delta\Delta C_t}$) with respect to the mock (DMSO-treated) and normalized for the housekeeping gene GAPDH. n=2, biological replicates, each with three technical replicates. The data presented are the mean of n = number of independent biological samples. Statistical significance was calculated using a Welch-corrected two-tailed t-test in GraphPad Prism; p-value: ns > 0.05, * \leq 0.05, ** \leq 0.01, *** \leq 0.001, **** \leq 0.0001.

7. G4 sequences: A minor but important point: A dedicated table should be provided listing the sequences of the G4s discussed, especially for non-classical G4s (e.g., HMGN1, IL17RA).

We have provided the sequences of the different structures targeted as requested by the reviewer. This is now illustrated in Table S8.

8. iM study (starting line 634): while the paragraph on iMs introduces novel findings, it raises several questions due to insufficient background and context on iM biology. Are the observed effects consistent with the putative roles of iMs? Why was iMab-based CUT&Tag (see 10.1093/nar/gkad626) not employed to further support the presented results, similar to the approach used for G4s? Is there a functional correlation between c-Myc's G4s and iMs? If so, how might this be explored? For example, would it be relevant to perform G4 CUT&Tag for ATENA with RVS experiments, followed by iM CUT&Tag for ATENA with PDS experiments? As previously noted, the authors must fully integrate the study of iMs by providing all necessary background and context, both in the introduction and in this dedicated section.

We agree with the reviewer that more background and discussion about iM's biology and their potential putative role should have been discussed to give more context to the results obtained. As stated in our answer to point 1, this has now been implemented. On the points raised: 1) we are on the same page with the reviewer - the responses observed upon iM-targeting might be

related to perturbation of the MYC-G4 dynamics, which is a fascinating relationship that needs further investigation. We have now added these aspects to the discussion of the revised manuscript (Line 1190-1192); 2) we also agree that implementing iMab CUT&Tag is a good approach to further assess the responses observed with ATENA. However, optimising CUT&Tag with a different antibody is not an easy task, and we feel this is significantly beyond the scope of the manuscript at this stage, especially considering that iM-targeting has been added as a response to the reviewers' comments.

9. General comment on the discussion: this quite long discussion recapitulates the main findings of this study, but fails in large parts to be a real discussion: for instance, the limitations of the techniques must be better described, the comparisons (advantages and drawbacks) with existing techniques must also be better discussed, etc. Some strong choices have been made all along this manuscript (for example to take BG4 as a representative G4 binding protein, or to use a single iM ligand) that must be better defended here. Also, a better contextualisation of these results in the global G4 and iM biology must be provided, insisting on unprecedented questions that are now open to investigations and must be addressed. Also, there is a strong bias in the discussion (lines 863-867): the cellular effects of ligands do not solely originate in their ability to modulate transcription but also, as discussed for instance with the aforementioned Durocher's paper, in triggering DNA damage in a G4-mediated manner. This is not an isolated example as many others could be cited including a Tarsounas' paper on PDS for instance (10.15252/emmm.202114501), or even one of the author's articles (10.1038/ncomms14432). Many other possibilities exist, as the cellular activity of G4 ligands (including PDS and PhenDC3) has also been linked to telomere instability, immunomodulation activity (e.g, 10.1093/nar/gkab500) and autophagy disruption (e.g, 10.1093/nar/gkz095), etc. Of course, G4-associated transcriptional control could be involved but many other molecular pathways, which remain to be fully understood, could be involved as well. The discussion should therefore adopt a more nuanced perspective to reflect this complexity.

We agree with the reviewer that our manuscript addresses several aspects that are challenging to concisely address in the discussion section, especially considering that this section is already quite lengthy on its own. Nevertheless, in the first round of revision, we have added a table to systematically compare all the available methods and contrast them with ATENA (Table S13), and we have now provided a more comprehensive overview of the responses elicited by G4-ligands beyond transcription (Lines 1234-1247). On the choice of BG4 and RVS, we feel that this is adequately justified in our manuscript. We clearly state that BG4 is merely a prototype for a G4 binding protein, and that the response may vary not only according to the ligand but also the protein binding to the G4. Nevertheless, BG4 is arguably the most characterised G4-binding antibody and is ideal to use for monitoring G4 occupancy through CUT&Tag. For RVS, we thoroughly mention the study from the Waller lab, where this peptide has been identified as the most selective for iM vs G4 (see the relevance of this in point 8), which is why we selected it for the study.

10. A final (personal) remark: in response to comment #38, the authors state that they find it inappropriate for a reviewer to use a non-peer-reviewed article to challenge their findings. First, they are entirely correct, and the reviewer should indeed apologize for this (notwithstanding the relative inelegance of the phrasing). Second, given the importance of Byrd's findings—now published in Nucleic Acids Research and thus available for use—this control was both critical and fully justified. As it turned out, the authors addressed

it, so all is well. Third, the request was not intended to "challenge" the findings but rather to ask for an additional control to further strengthen them—for the sake of the story

Apology accepted. We feel it is important to flag these issues for the sake of both the editor, who relies on the professionalism of the reviewers, and of the young scientists co-authoring this paper, who should be made aware of what is ethical and what is not when reviewing manuscripts, and quoting preprints to challenge findings of a submitted manuscript is clearly crossing the line. We agree that the controls requested were critical, which is why they were included in the initial submission to Nat. Cell Biol. and were not added as a response to comment #38, as the reviewer implies. Additionally, as per our original response, we performed a CUT&Tag-based enrichment (qPCR); therefore, the G4-unspecific tagmentation reported by Byrd would not have affected our findings regardless of these controls (see our response #38 from the original submission), which is why we were disappointed by this comment in the broadest possible sense. We hope that this can be useful to the reviewer in their future activities and, as stated above, we accept their apologies.

Reviewer 2

11. OK. Please still present it as one of the key limitations when using this technology in discussion. The PAM site requirement may restrict many gene to be considered, and other targeting approach may be more widely accessible instead, e.g. small molecule, peptide, ASO, aptamer, nanobody, etc.

We agree with the reviewer that PAM site requirements represent an important limitation of ATENA that deserves explicit discussion. We have now added this to our discussion section, acknowledging that PAM availability can constrain target selection for some genes of interest (Line 1355-1357). However, it is also important to put this limitation into perspective. Our analysis suggests that the 20-60 bp effective range for dCas9 positioning, combined with the relatively frequent occurrence of PAM sequences, provides reasonable flexibility for most genomic targets. Additionally, the expanding toolkit of Cas variants with different PAM requirements can further broaden targeting options, as we clearly state in the revised manuscript.

We acknowledge that alternative approaches such as small molecules, peptides, ASOs, aptamers, and nanobodies each offer distinct advantages (as reported in Table S13). We consider ATENA not as a replacement for these approaches, but rather as a complementary tool that offers unique advantages for mechanistic studies (precise spatial control and the ability to deliver diverse cargos to specific sites). Nevertheless, we have expanded our discussion to explicitly highlight the limitations of ATENA as requested (Line 1355-1365).

12. OK. One potential future direction will be then to use other dCas protein with different more promiscuous PAM requirement, or some other proteins with no PAM requirement. The author may consider adding to the discussion/future directions if applicable.

We appreciate the reviewer highlighting this crucial future direction. We have now added a discussion of alternative targeting systems that could expand ATENA's applicability beyond the current PAM constraints as suggested, see also response 11.

13. OK. A follow up question is that for this opposing effect of G4 ligands, are they also dose-dependent, or the effect only diverge when the dose become much more excess than the G4 target or G4 binding proteins in cells?

This is an important point. In all our experiments, the ligand concentrations we use are determined by the CAPA assay, which measures the extent of ligand engagement with dCas9-Halo. We specifically choose concentrations that achieve comparable levels of dCas9-ligand conjugation across all ligands tested to avoid the presence of free ligands in the cellular context. When we compare the effects of C1-PyPDS₂ and C1-PhenDC₃ at MYC-G4 versus PVT1-G4, we are using functionally equivalent doses in terms of actual ligand-protein engagement and, most importantly, never in excess with respect to the G4-target.

Therefore, the opposing effects we observe - both ligands causing MYC downregulation but having opposite effects at the PVT1 locus - occur at comparable ligand concentrations. This strongly supports our conclusion that these differences arise from the distinct interaction modes of each ligand with the specific G4-protein complexes at each site, rather than from dose-related artefacts or differences in ligand availability. Additionally, these responses can be recapitulated also using free ligands (not tethered to dCas9), further supporting our interpretation.

14. While the bindings between free ligand and chloroalkane-modified ligand to target of interest are assessed, it is important to also verify the binding of dCas9-Halo-ligand with the target of interest, as dCas9 is large in size, and may affect the binding due to steric effects.

The reviewer raises a valid technical concern here. Indeed, we provide several lines of evidence that support effective G4 engagement by the complete system. For instance, BG4 CUT&Tag experiments (Fig. 2f, 5c) demonstrate that ATENA treatment specifically alters G4-protein interactions (BG4) at targeted sites - C1-PyPDS₂ enhances BG4 binding at PVT1-G4 whilst both ligands reduce it at MYC-G4. These changes in G4 accessibility would be difficult to explain without effective ligand-G4 interaction occurring in the cellular context.

Additionally, the distance-dependent effects we observe (Fig. 2e) support a model where the ligand must physically reach and engage the G4 structure. sgRNAs positioned too far from the target G4 (>60 bp) show no transcriptional effects, whilst those within the effective range show robust responses, consistent with the ligand successfully bridging the distance from dCas9 to the G4. Finally, our FRET-based evidence in vitro clearly supports the notion that the ligands can still engage with G4s in vitro, also when tethered to the bulky dCas9.

15. The authors mentioned that the C-terminal domain of the Cas9 protein points towards the 3' end of the PAM sequence, which means the ligands will be pointed away from the G4 structure when using NT-sgRNA (as shown in fig. 1g). Given this, please clarify why NT-sgRNA_{FRET-21}-C1-PDS₂ also causes an increase in FRET signal? Please explain it more explicitly. For future, are there any general design principles that the author can summarize for the reader in the discussion.

When using NT-sgRNA, the C-terminal domain (where Halo-ligand is attached) points away from the G4 structure based on the crystal structure orientation, as correctly pointed out by the reviewer. However, we believe that the proximity of the dCas9-Halo/ligand complex to the G4 target (just 21 bp), combined with the flexibility of the PEG-linker, allows for some engagement despite the directional mismatch. Indeed, NT-sgRNA_{FRET-21}- C1-PyPDS₂ shows

FRET signal increase, but to a lesser extent than the optimally oriented sgRNAs, supporting this notion. Therefore, for cellular application, we suggest prioritising sgRNAs where the dCas9 C-terminus points toward the target G4, which we have now explicitly spelled out in the manuscript and in the discussion part, where we summarise the design principles.

16. For Fig. 1h, please include the Δ FRET of NT-sgRNA_{FRET-41}.

The Δ FRET value for NT-sgRNA_{FRET-41} is already included in Figure 1h. As shown, the Δ FRET efficiency for NT-sgRNA_{FRET-41} with Cl-PyPDS₂ treatment is zero, indicating no detectable ligand-induced FRET change.

17. The authors demonstrated that targeting MYC-iM has an opposite effect compared to targeting MYC-G4. However, the underlying mechanism remains unclear – is this due to the characteristics of RVS-pep, or the different properties of iM from G4? It will be good to perform a melting test on RVS-pep to verify whether it stabilizes or destabilizes the iM.

We thank the reviewer for this insightful comment. To address this, we performed circular dichroism (CD) melting experiments to determine whether Cl-pep-RVS₄ stabilises or destabilises the i-motif-forming structure of the oligo sequence c-MYC-C52.

CD melting revealed that Cl-pep-RVS₄ stabilises the i-motif structure, with 10 equivalents of Cl-pep-RVS₄ increasing the melting temperature of $\sim 6^{\circ}\text{C}$ ($p < 0.01$). This stabilisation effect demonstrates that Cl-pep-RVS₄ directly binds to and stabilises the MYC i-motif structure, causing transcriptional stimulation, as described in the literature. This data is now reported in Extended Data Fig. 6 and Extended Data Table 5, along with a detailed discussion on the mechanism and state of the art, as mentioned in responses 1 and 9 to reviewer 1.

18. Since MYC-iM is located on the opposite strand of MYC-G4, it will also be good to assess with sgRNA_{MYC+22}, which is on the same strand as the iM (NT strand).

We appreciate the reviewer's suggestion to assess sgRNA_{MYC+22}, which targets the same strand bearing the i-motif. We have indeed performed these experiments (see Figure R2), but the results cannot be reliably attributed to i-motif targeting. The sgRNA_{MYC+22} target region is in proximity to critical promoter regions (TATA-box), which introduces significant confounding variables, as we demonstrate thoroughly in our manuscript, leading to interference with transcriptional machinery binding sites, promoter accessibility, or other regulatory elements, rather than being specifically attributable to i-motif modulation.

We acknowledge that this represents a limitation in our current experimental approach, and future studies with more distally located guide RNAs or alternative experimental designs may be needed to overcome this (see response 12), but we hope that the reviewer can appreciate that we have performed this experiment, although it cannot be used for the final manuscript.

Figure R2 RT-qPCR of the indicated genes in MCF7 cells stably expressing dCas9-Halo transfected with sgRNA_{MYC-19} sgRNA_{MYC+22} or sgRNA NTC and incubated for 48h in the presence of 10 μ M of Cl-pep-RVS₄ or DMSO (mock). The expression values are represented as fold change ($2^{-\Delta\Delta C_t}$) with respect to the mock (DMSO-treated) transfected samples and after normalization for the housekeeping gene (GAPDH). n=2, biological replicates, each with three technical replicates. The data presented are the mean of n = number of independent biological samples. Statistical significance was calculated using a Welch-corrected two-tailed t-test in GraphPad Prism; p-value: ns > 0.05, * \leq 0.05, ** \leq 0.01, *** \leq 0.001, **** \leq 0.0001.

19. Fig. 5C shows a non-significant increase in IL17RA expression levels. It might be good to further support the hypothesis that IL17RA is unaffected by using more sgRNAs or other ligands.

We thank the reviewer for this valuable suggestion. To further support our hypothesis that *IL17RA* expression remains unaffected by G-quadruplex targeting, we conducted additional experiments employing both alternative sgRNAs and additional ligands. We designed a novel sgRNA targeting a downstream region of *IL17RA* to confirm that the observed effects were not attributable to the specific targeting location of our original sgRNA. In accordance with our initial findings, this alternative sgRNA similarly demonstrated no significant effect on *IL17RA* expression levels (Extended Data Fig. 7 a,b), thereby supporting our interpretation. Furthermore, we examined the responses elicited by other chloroalkane engineered ligands used in the manuscript Cl-PhenDC3₂ (Extended Data Fig. 7, a) and Cl-PDC₂ (Extended Data Fig. 7, b). Both Cl-PhenDC3₂ and Cl-PDC₂ treatment failed to trigger significant alterations in *IL17RA* expression levels. This complementary evidence reinforce the notion that baseline transcriptional levels of G4-containing genes can serve as predictive indicators for the magnitude of transcriptional perturbation following ligand treatment.

a

b

Extended data Figure 7 *IL17RA* expression is not affected when targeting the G4 located in its promoter. RT-qPCR for *IL17RA* expression in MCF7 cells stably expressing dCas9-Halo, transfected with either sgRNA_{IL17RA-20}, sgRNA_{IL17RA+36}, or sgRNA NTC and treated with (2.5 μ M) CI-PhenDC3₂ (a) or CI-PDC₂ (b) or 48h after transfection. The expression values are represented as fold change ($2^{-\Delta\Delta C_t}$) with respect to the mock (DMSO-treated) and normalized for the housekeeping gene GAPDH; n=2, biological replicates, each of which includes two technical replicates. Data presented are the mean of n = number of independent biological samples. Statistical significance was calculated using a Welch-corrected two-tailed t-test in GraphPad Prism; p-value: ns > 0.05, * ≤ 0.05 , ** ≤ 0.01 , *** ≤ 0.001 , **** ≤ 0.0001 .

20. To further distinguish this work with previous work (PMID: 38961283), please clarify the major differences between the two studies. This clarification will help highlight the novel contributions and significance of the current research within the context of existing literature. The authors should include it as supplementary note.

We have already included a supplementary table (S13) to highlight the main differences between our approach and the one mentioned by the reviewer, as well as additional non-CRISPR-based methods. Nevertheless, we have further highlighted technical differences between the two methods to ensure this point is addressed (Line 1079-1112).

21. The authors used PyPDS in their analogues but named them Cl-PDSn. To avoid confusion, please rename them as Cl-PyPDSn.

We have re-named the analogues accordingly as suggested.

22. Please clarify what NTC refers to in this context, whether it is sgRNA targeting unrelated sequences or scrambled sgRNA.

We have clarified that this is a scrambled sgRNA control as requested.

23. Make sure that the significance levels are displayed for all samples in the graphs, including NTC.

We have checked the whole document and ensured that significance levels are consistently displayed throughout the graphs.

24. In page 11 line 287, please specify which SI figure it is after T-sgRNAFRET-41.

We have specified the SI figure number.

25. For clearer illustration, label the 5' and 3' end of the sequences in Fig. 1g.

We have edited the figure as requested.

26. In page 27 line 652 and page 30 line 730, please confirm whether it is chloroalkene or chloroalkane side chain.

We apologise for the typo and we have now corrected the spelling.

Reviewer 1

1. Globally speaking, this is clearly a beautiful piece of science that deserves to be published in Nat. Commun.

We thank the reviewer for their words and for deeming our revised manuscript suitable for publication

Reviewer 2

2. The authors should try to make the manuscript more concise/organized to improve the readability, despite the amount of newly added data during rounds of revision. This will make the messages clearer, and more people will be able to understand the main takeaways and attempt to use their technology.

We have tried our best to improve conciseness and readability in the final version submitted. However, there is a substantial number of additional experiments that we had to introduce during the revision stages and this clearly reflects on the density of the final manuscript.